# Deep Gaussian from Motion: Exploring 3D Geometric Foundation Models for Gaussian Splatting

**Yu Chen**[1]    **Rolandos Alexandros Potamias**[2]    **Evangelos Ververas**[2]
**Jifei Song**[3]    **Jiankang Deng**[2]    **Gim Hee Lee**[1]
[1]National University of Singapore    [2]Imperial College London    [3]University of Surrey

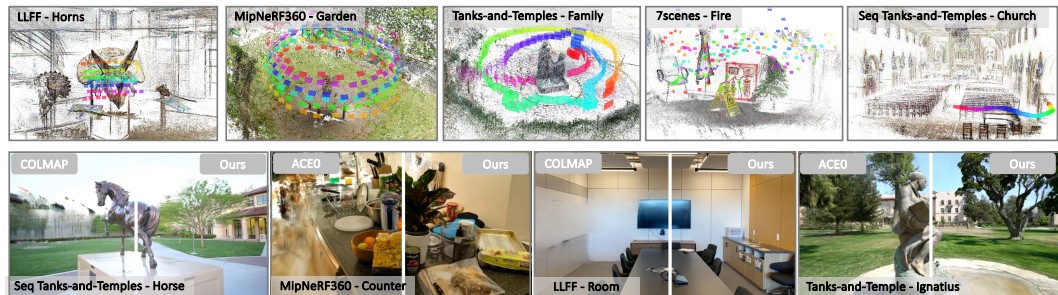

Figure 1: **Reconstruction results of DeepGfM**. Our method reconstructs scenes from unordered image collections without COLMAP poses. Top row: Reconstructed camera poses and point clouds. Bottom row: Rendered novel views compared with COLMAP and ACE0.

## Abstract

Neural radiance fields (NeRF) and 3D Gaussian Splatting (3DGS) are popular techniques to reconstruct and render photorealistic images. However, the prerequisite of running Structure-from-Motion (SfM) to get camera poses limits their completeness. Although previous methods can reconstruct a few unposed images, they are not applicable when images are unordered or densely captured. In this work, we propose a method to train 3DGS from unposed images. Our method leverages a pre-trained 3D geometric foundation model as the neural scene representation. Since the accuracy of the predicted pointmaps does not suffice for accurate image registration and high-fidelity image rendering, we propose to mitigate the issue by initializing and fine-tuning the pre-trained model from a seed image. The images are then progressively registered and added to the training buffer, which is used to train the model further. We also propose to refine the camera poses and pointmaps by minimizing a point-to-camera ray consistency loss across multiple views. When evaluated on diverse challenging datasets, our method outperforms state-of-the-art pose-free NeRF/3DGS methods in terms of both camera pose accuracy and novel view synthesis, and even renders higher fidelity images than 3DGS trained with COLMAP poses. Our project page is available at https://aibluefisher.github.io/DeepGfM.

## 1  Introduction

In recent years, the rapid advancement of 3D reconstruction techniques has enabled diverse applications for digitizing real-world scenes. Travelers and tourists routinely capture photographs and videos of landmarks, which online platforms now process using neural radiance fields (NeRF) [42] or 3D Gaussian Splatting (3DGS) [26] to generate photorealistic 3D models.

39th Conference on Neural Information Processing Systems (NeurIPS 2025).

Despite their fidelity, NeRF and 3DGS pipelines rely heavily on classical Structure-from-Motion (SfM) methods like COLMAP [47] for precomputing camera poses. This decoupled setup introduces inconsistencies: SfM relies on sparse keypoint reprojection errors, while neural rendering optimizes dense photometric losses. Even minor pose inaccuracies can cause ghosting artifacts, degrading visual quality (*cf.* Bottom row of Fig. 1).

This limitation raises a fundamental question: *Can we train a fully differentiable reconstruction network guided solely by photometric loss?* While recent work like CF-3DGS [19] avoids explicit pose estimation, it relies on consecutive-frame poses and monocular depth priors, limiting generalization to unordered images or large camera motions. Thus, a unified framework for joint geometry estimation and rendering is critical for robust 3D reconstruction across diverse inputs.

The 3D geometric foundation model DUSt3R [62] marked a milestone toward end-to-end neural SfM, inspiring pose-free 3DGS methods [49, 69]. However, these approaches handle only image pairs due to DUSt3R's limitations. Extensions like Spann3R [58] and its variants (VGGT [59], Fast3R [68], and FLARE [73]) fine-tuned DUSt3R on globally aligned pointmaps, thus extending the input of DUSt3R from just image pairs to entire image sequences. However, their geometric accuracy lags behind traditional SfM. Moreover, the GPU memory requirement of these methods grows rapidly when handling hundreds of images(*cf.* Table 1), hindering high-fidelity rendering on consumer hardware. This motivates our work to fine-tune existing pre-trained DUSt3R variants [58, 68, 73, 59] for high-fidelity novel view synthesis from videos or unordered image collections.

We propose **Deep Gaussian from Motion (DeepGfM)**, an end-to-end neural reconstruction framework that unifies geometry estimation and novel view synthesis for video sequences and unordered image collections. Our approach leverages a pre-trained 3D geometric foundation model (*e.g.*, Spann3R or DUSt3R variants) as a scene regressor, extending its output beyond pointmaps to 3D Gaussian primitives – enabling direct optimization of scene geometry and appearance via photometric losses. To address the limitations of prior pose-free methods, we introduce a **progressive training strategy** with three core stages: 1) **Initialization**: The scene regressor processes images from a training buffer, predicting pointmaps and 3D Gaussians in a global coordinate frame. We then compute initial camera poses via RANSAC and PnP. 2) **Iterative Refinement**: To mitigate pose inaccuracies, we propose a point-to-camera ray consistency loss, jointly refining camera poses and pointmaps. Newly registered images are added to the training buffer, and the scene regressor is fine-tuned on this expanded set using photometric losses. This loop continues until no further images can be registered. 3) **Final Optimization**: Accumulated errors are alleviated by re-optimizing all camera poses with the previous results as initialization, ensuring global consistency. Our framework is architecture-agnostic: it retains the internal structure of the pre-trained regressor (*e.g.*, DUSt3R's transformer backbone) while adapting its output space for Gaussian primitives. This design balances generalization (inheriting geometric priors from foundation models) and flexibility (supporting diverse DUSt3R variants).

Experiments demonstrate that DeepGfM surpasses both pose-free NeRF/3DGS methods and COLMAP-based 3DGS in pose accuracy and rendering quality. While feed-forward pipelines like VGGT and Fast3R prioritize speed, DeepGfM addresses the reconstruction of dense image capture workflows to enable high-fidelity novel view synthesis without reliance on pose annotations or large-scale memory-intensive methods.

Our main contributions are summarized as follows:

- **Pose-free Foundation Model Adaptation**. Unlike VGGT/MegaSAM, which rely on pre-computed poses (potentially affected by inaccuracies), our pipeline operates without pose annotations. This is achieved by refining Gaussian geometries dynamically to align photometric appearance with ray-consistent novel view synthesis.

- **Progressive and Modular Framework Design**. The progressive design enables iterative scalability, addressing GPU bottlenecks evident in VGGT-like pipelines. Modularity ensures robustness against scene diversity, allowing refinement of components independent of memory constraints imposed by dense image sets.

- **Scene-specific Gaussian Prediction**. Our method dynamically predicts Gaussian geometries for each input scene, adapting to its unique photometric and geometric characteristics for high-quality synthesis – a flexibility less evident in feed-forward methods like VGGT/MegaSAM, which process inputs less adaptively.

Table 1: GPU memory usage (GB) of neural scene reconstruction methods with 200 images as input.

| Method | CF-3DGS [19] | DUSt3R [62] | Fast3R [68] | VGGT [59] | Ours |
|---|---|---|---|---|---|
| GPU Memory | OOM | OOM | 33.17 | 40.63 | 23.07 |

## 2 Related Work

**Neural Rendering.** Neural radiance fields [42] enable rendering from novel viewpoints with encoded frequency features [54]. Many follow-up works try to improve the rendering and training efficiency [34, 71, 18, 10] by encoding scenes into sparse voxels, multi-resolution hash tables [43], or three orthogonal axes and planes [10]. Another branch of NeRF methods focuses on generalizable NeRF [60, 11, 24, 52, 36], alleviating the aliasing [2, 4]. , registering multiple blocks of NeRF [20, 13], and extending NeRF to city-scale scenes [53, 55, 45, 40, 65]. Different from NeRF, 3D Gaussian Splatting [26] (3DGS) initializes 3D Gaussians from a sparse point cloud and renders scenes by differentiable rasterization, and can achieve real-time rendering performance. Follow-up works include learning the scene geometry implicitly by MLP [37] or GNN [57], fitting the surface via 2D Gaussians [23, 21], alleviating the aliasing issue [72, 67, 31, 50], improving the training efficiency [33, 27, 35, 14] for large-scale scenes.

**Pose-Free Neural Rendering.** To alleviate the reliance of SfM poses, NeRFmm [64] jointly optimizes NeRF and camera pose embeddings. BARF [32] proposes joint training of NeRF with imperfect camera poses from coarse-to-fine, where high-frequencies are progressively activated during training to alleviate the gradient inconsistency issue. GARF [15] extends BARF with a Gaussian activation, enabling training a positional-embedding less coordinate network. However, these methods can only handle forward-facing cameras or require accurate pose priors to converge. Nope-NeRF [5] and CF-3DGS [19] leverage the inter-frame relationship (*e.g.* relative poses) and monocular depth maps as regularization to train NeRF or 3DGS without precomputed camera poses, but limited to short image sequences. Moreover, Nope-NeRF and CF-3DGS are highly susceptible to failure when camera poses change significantly or images are unordered. InstantSplat [17] leverages the pre-trained DUSt3R [62] to regress dense pointmaps between image pairs, followed by computing camera poses by aligning the pointmaps into a global coordinate frame. However, aligning dense pointmaps is time- and memory-consuming. As a result, InstantSplat can only handle a very few images. Other related work includes Splatt3R [49] and NoPoSplat [69] to predict 3D Gaussians from pairwise unposed images. Other methods include introducing the SLAM pipeline into the NeRF [51, 74] or 3DGS [25, 39, 66] for indoor video sequences.

**Neural 3D Reconstruction.** DUSt3R [62, 30] reconstructs dense pointmaps from image pairs. By aligning dense points, DUSt3R can obtain globally aligned camera poses and dense points. Though DUSt3R generalizes very well since it is trained on massive diverse datasets, it produces only local points and requires long post-processing time with a heavy GPU memory burden to obtain globally aligned results. Spann3R [58] is the first work that tries to eliminate the post-processing step of DUSt3R by utilizing a spatial-memory to decode the global points. Concurrent to our work, methods [68, 73] improve the reconstruction speed and generality by fine-tuning DUSt3R on globally aligned 3D points. ACEZero [9] reconstructs from unordered image collections via MLP and achieves faster reconstruction speed than COLMAP while with less accurate camera poses. Orthogonal to existing methods, our work *focuses on the reconstruction accuracy in terms of both camera pose and novel view synthesis*.

## 3 Method

We first give the preliminaries of our scene regressor network in Sec. 3.1 and 3.2, followed by introducing our network architecture and progressive training strategy in Sec. 3.3. Fig. 2 shows a brief illustration of our network.

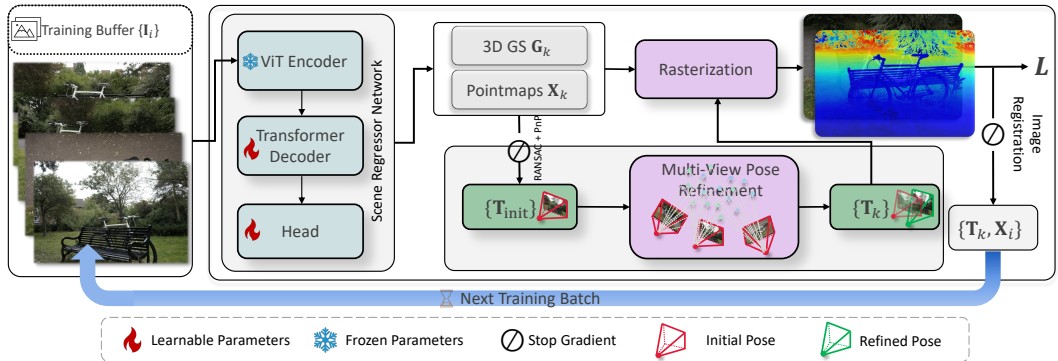

Figure 2: **The training pipeline of our proposed method**. The method follows the classical incremental SfM reconstruction pipeline with the key difference that the input is no longer an image but a pair of images in a progressively updated training buffer. The scene regressor network is trained as follows: 1) Use Spann3R [58] as the scene regressor network to predict 3D Gaussians $\mathbf{G}_k$ and pointmaps $\mathbf{X}_k$ from a pair of images. 2) Leverage RANSAC and a PnP solver to obtain the initial camera poses based on direct 2D-3D correspondences. 3) Refine the coarse camera poses by minimizing the point-to-ray consistency loss between 3D tracks and camera centers. 4) Rasterize the 3D Gaussians with the refined camera poses to render images. A photometric loss is adopted for back-propagating gradients. 5) After each training epoch, we update the training buffer by registering more images.

## 3.1 Preliminaries

Our network architecture is based on DUSt3R [62] and Spann3R [58]. Given an image pair $(\mathbf{I}_i, \mathbf{I}_j)$, DUSt3R predicts the corresponding pointmaps $(\{\mathbf{X}^{i,i}\}, \{\mathbf{X}^{j,i}\})$ for each image, where $\mathbf{X}^{j,i}$ denotes the pointmap $\mathbf{X}_j$ expressed in camera $i$'s coordinate frame.

**DUSt3R** uses a ViT [16] as a shared encoder for both images and two transformer decoders for the reference image $i$ and the target image $j$, respectively. The two decoders denoted as *reference decoder* $\mathcal{D}_{\text{ref}}$ and *target decoder* $\mathcal{D}_{\text{tgt}}$ consist of two projection heads $\mathcal{H}_{\text{ref}}, \mathcal{H}_{\text{tgt}}$ that map the decoded features into pointmaps:

$$\mathbf{f}_i^{\text{e}}, \mathbf{f}_j^{\text{e}} = \mathcal{V}(\mathbf{I}_i, \mathbf{I}_j), \mathbf{f}_i^{\text{d}} = \mathcal{D}_{\text{ref}}(\mathbf{f}_i^{\text{e}}, \mathbf{f}_j^{\text{e}}), \mathbf{f}_j^{\text{d}} = \mathcal{D}_{\text{tgt}}(\mathbf{f}_j^{\text{e}}, \mathbf{f}_i^{\text{e}}), \tag{1}$$
$$\mathbf{X}^{i,i}, \ \mathbf{X}^{j,i} = \mathcal{H}_{\text{ref}}(\mathbf{f}_i^{\text{d}}), \ \mathcal{H}_{\text{tgt}}(\mathbf{f}_j^{\text{d}}).$$

DUSt3R reconstructs image pairs in a local coordinate frame. When handling more than two images, DUSt3R uses a post-processing step to align the pairwise dense pointmaps to a global coordinate frame, which is time-consuming and can exceed the GPU memory limitation.

**Spann3R** proposes a feature fusion mechanism to predict pointmaps $(\{\mathbf{X}^{i,g}\}, \{\mathbf{X}^{j,g}\})$ in a global coordinate frame. It computes a fused feature $\mathbf{f}_t^G$ in the $t$-th training epoch from a spatial feature memory. The reference decoder inputs the fused feature for reconstruction, and the target decoder produces features for memory querying. Furthermore, Spann3R uses two additional projection heads to compute the key and query feature for reconstructing the next image pairs, and a memory encoder $\mathcal{V}_{\text{mem}}$ which encodes the pointmaps from the reference decoder:

$$\mathbf{f}_j^Q = \mathcal{H}_{\text{tgt}}^Q(\mathbf{f}_j^d), \ \mathbf{f}_i^K = \mathcal{H}_{\text{ref}}^K(\mathbf{f}_i^d), \ \mathbf{f}_i^V = \mathcal{V}_{\text{mem}}(\mathbf{X}^{i,g}). \tag{2}$$

Although Spann3R can reconstruct out-of-distribution scenes, it reconstructs images individually and is limited to very short frames due to GPU memory limitations. Moreover, the 3D points predicted by Spann3R in these scenes lack accuracy. We refer readers to [62, 58] for more details.

## 3.2 Neural Scene Representation

Modern 3D scene representations fall into two categories: explicit (*e.g.*, 3D Gaussian primitives [26]) and implicit (neural networks [37, 46]). Our work builds on Spann3R [58], extending it to predict both dense pointmaps and 3D Gaussian primitives. We refer to Spann3R as the *scene regressor network* $f_{\text{SCR}}$, which analogs to the scene regressor in pose regression or localization networks [6, 9].

Unlike the sparse scene regressor that takes an image as input, our scene regressor takes an image pair as input and predicts dense pointmaps $\mathcal{X} = \{\mathbf{X}_i\}$ and per-pixel 3D Gaussians $\mathcal{G} = \{\mathbf{G}_i\}$ in a global coordinate frame:

$$(\mathcal{X}_i, \mathcal{G}_i;\ \mathcal{X}_j, \mathcal{G}_j) = f_{\mathrm{SCR}}(\mathbf{I}_i, \mathbf{I}_j), \tag{3}$$

where $\mathbf{I}_i$ is the reference and $\mathbf{I}_j$ is the target image.

By rasterizing the set of 3D Gaussian primitives $\mathcal{G}$, we back-propagate gradients to the model using a photometric loss. More specifically, a 3D Gaussian primitive is composed of the opacity $o$, the mean $\mathbf{u}$, the covariance $\mathbf{\Sigma}$, and the coefficients of the spherical harmonics $\mathbf{SH}$. The covariance is decomposed into a rotation matrix $\mathbf{R}$ and a scaling matrix $\mathbf{S}$ to ensure the positive semi-definiteness: $\mathbf{\Sigma}_i = \mathbf{R}\mathbf{S}\mathbf{S}^\top\mathbf{R}^\top$. In addition, instead directly predicting the mean $\mathbf{u}$ for each 3D Gaussian, we predict an offset $\Delta\mathbf{X}$ and apply it to the pointmaps to obtain the mean $\mathbf{u} = \mathbf{X} + \Delta\mathbf{X}$. To render the color for a pixel $\mathbf{p}$, the 3D Gaussians are projected into the image space for alpha blending with the projected 2D Gaussian $\mathbf{G}^{\mathrm{proj}}$:

$$\mathbf{C} = \sum_i \mathbf{c}_i \alpha_i \prod_{j=1}^{i-1}(1 - \alpha_j), \tag{4}$$

where $\alpha_i$ is the rendering opacity and is computed by $\alpha = o \cdot \mathbf{G}^{\mathrm{proj}}(\mathbf{p})$, $\mathbf{c}_i$ is the per-pixel color that computed from the spherical harmonics $\mathbf{SH}$. In practice, Eq. (4) is computed using a differentiable rasterizer [26]:

$$\hat{\mathbf{I}} = \mathcal{R}(\mathbf{T}, \mathbf{K}; o, \mathbf{u}, \mathbf{R}, \mathbf{S}, \mathbf{SH}) = \mathcal{R}(\mathbf{T}, \mathbf{K}; \{\mathbf{G}_i\}), \tag{5}$$

where $\mathbf{T}$ is the camera extrinsics and $\mathbf{K}$ is the intrinsics.

### 3.3 Progressive Neural Reconstruction

Fine-tuning a pre-trained geometric foundation model such as DUSt3R on unseen scenes is challenging. This is because they need ground-truth 3D data as supervision. However, obtaining ground-truth 3D points is difficult and expensive. We circumvent this limitation through a self-supervised progressive approach.

#### 3.3.1 Seed Initialization

Given an unordered image set $\mathcal{I} = \{\mathbf{I}_i\}$, we first select a *seed image* for initialization. This differs from incremental SfM, which requires a *seed image pair* with sufficient matching inliers and a wide baseline for two-view reconstruction. We follow a similar criterion to search for an initial image that overlaps with as many other images as possible for faster convergence. To stabilize the training, we use NetVLAD [1] to compute a global descriptor for each image. Then we compute similarity scores between image descriptors. We further build a similarity graph, where the node represents an image, the edge represents an image pair, and the edge weight represents the similarity score of that pair. We remove an edge if its weight is lower than the threshold $s_{\mathrm{sim}}$. We then select the node that has the maximum degree as the seed image $\mathbf{I}_{\mathrm{seed}}$. Intuitively, a node with a maximum degree suggests it has the most adjacent images, which is beneficial for image registration. After selecting the seed image, we finetune the scene regressor in a self-supervised manner. Specifically, we set the seed image as the reference frame with identity camera pose $\mathbf{T}_{\mathrm{seed}} = \mathbf{I}$. We then compute a photometric loss:

$$\mathcal{L}_{\mathrm{photo}} = \sum \|\mathbf{I} - \hat{\mathbf{I}}\|_1 = \sum \|\mathbf{I} - \mathcal{R}(\mathbf{T}_{\mathrm{seed}}, \mathbf{K}, \{\mathbf{G}_i\})\|_1. \tag{6}$$

Note that, during initialization, the seed image serves as both the reference and target image to the scene regressor, and the camera pose is fixed as an identity matrix.

#### 3.3.2 Progressive Registration

After seed initialization, we progressively increase the training buffer by registering images $\mathcal{I}_{\mathrm{buf}} = \{\mathbf{I}_k\}$ in a training epoch. The image registration stage includes: 1) a coarse camera pose estimation step to obtain initial camera poses with the fine-tuned scene regressor. 2) a camera pose refinement step to improve the accuracy of registered cameras. With newly registered images, we further finetune the scene regressor in the current epoch. Upon training convergence, we increase the training buffer by selecting more images. This process is repeated until all images are registered.

**Coarse Camera Pose Estimation.** Given a registered reference image $\mathbf{I}_{\text{ref}}$ and a unregistered target image $\mathbf{I}_k$, we pass them to the scene regressor and obtain the 3D points $\{\mathbf{X}_k\}$ in a global coordinate frame. Since we have the coordinates $\{\mathbf{y}_k\}$ of each image pixel and their corresponding 3D coordinates $\{\mathbf{X}_k\}$, we can easily find the 2D-3D correspondences $\{(\mathbf{y}_k, \mathbf{X}_k)\}$. We then use RANSAC and a PnP solver to obtain a coarse camera pose:

$$\mathbf{T}_k^{\text{coarse}}, S_k = \text{PnP}(\mathbf{K}, \{(\mathbf{y}_k, \mathbf{X}_k)\}), \tag{7}$$

where $S_k$ is the number of inliers and $\mathbf{X}_k = f_{\text{SCR}}(\mathbf{I}_{\text{ref}}, \mathbf{I}_k)$. $\mathbf{I}_{\text{ref}}$ is the reference image and $\mathbf{I}_k$ is the target image we want to register. We add the target image $\mathbf{I}_k$ into the training buffer only when the inlier number is larger than the inlier threshold $s_{\text{inlier}}$. After initialization, the seed image is selected as the reference image. In the following training batches, we select the reference image from the registered images that connect to most of the unregistered images.

**Multi-View Consistent Pose Refinement.** From the previous steps, the camera poses of newly registered images can still be inaccurate. This is because: 1) some regions in images have not been observed by the scene regressor; 2) each target image is registered individually, which lacks multiple view constraints. Although ACE0 [9] uses an MLP pose refiner to alleviate this problem during training, we experimentally found that it does not improve pose accuracy with our transformer-based scene regressor. This is because ACE0 uses MLP as the scene coordinate decoder, and each pixel is individually mapped onto the 3D space. ACE0 thus enables network training by mixing millions of pixels from mutiple views in a training batch, and the multiple-view constraint helps constrain the network training. However, since we use a transformer-based decoder and due to the GPU memory limitation, we can use only a limited number of views in each training batch, which can easily diverge the network training. To solve the aforementioned issue, we propose to further refine the coarse camera poses by minimizing a point-to-camera ray consistency loss which involves multiple views:

$$\mathbf{X}_i, \mathbf{T}_k^{\text{refine}} = \underset{\mathbf{X}_i, \mathbf{R}_k, \mathbf{t}_k}{\arg\min} \sum_{i,k} \rho(\|d_{i,k} \cdot \boldsymbol{\nu}_{i,k} - (\mathbf{X}_i - \mathbf{t}_k)\|_2), \tag{8}$$

where $\mathbf{t}_k$ is the camera position for image $\mathbf{I}_k$, $d_{i,k}$ is the scaling factor between a 3D point $\mathbf{X}_i$ and the camera position $\mathbf{t}_k$, $\boldsymbol{\nu}_{i,k}$ is the ray direction between $\mathbf{X}_i$ and $\mathbf{t}_k$. This loss function is more robust than the reprojection loss in ACE0 [9] since the error in Eq. 8 is bounded [75]. And since the scene regressor already provided a coarse but reliable initial estimation, the convergence is fast. During optimization, we fix the camera pose of the seed image for the gauge ambiguity and fix the scaling factor between the seed image and its most similar adjacent image for the scale ambiguity. Moreover, for a new training epoch, we fix the camera poses registered in the previous epoch and only optimize the camera poses registered in the current training epoch to improve the optimization efficiency. Fixing camera poses registered from previous epochs is important for our method to reconstruct scenes at a larger scale because:

- After the previous fine-tuning epochs, the camera poses and pointmaps are accurate enough.
- By optimizing only local parameters, the GPU memory footprint is highly reduced. As a comparison, the global alignment step of DUSt3R and InstantSplat will encounter the OOM issue even with an 80GB A100 GPU.

After obtaining the refined camera poses, we fine-tune the scene regressor with a minor modification to Eq. (6) by:

$$\mathcal{L}_{\text{photo}} = \sum_k \|\mathbf{I} - \mathcal{R}(\mathbf{T}_k^{\text{refine}}, \mathbf{K}, \{\mathbf{G}_i\})\|_1. \tag{9}$$

### 3.3.3 Finalizing Neural Scene Reconstruction

We propose a two-stage strategy to improve the final reconstruction quality when all images have been registered or no more images can be added to the training buffer. The first stage is to optimize all camera poses using Eq. (8). This is because we incrementally register images and errors accumulate during training. In this stage, we only fix the camera pose of the seed image, and camera poses obtained from all previous training epochs are used as initial values for optimization. Since the initial values are accurate enough, the final optimization converges very fast. To further improve the image rendering quality, we proposed to refine the scene details using explicit 3D Gaussian primitives [26] in a second stage. This is because we only used fixed low-resolution images during the training of our scene regressor due to GPU memory limitations. The scene regressor can therefore only represent the coarse scene geometry. In the second stage, we use the same strategy as in [26] for 3D Gaussian densification and pruning during refinement.

# 4 Experiments

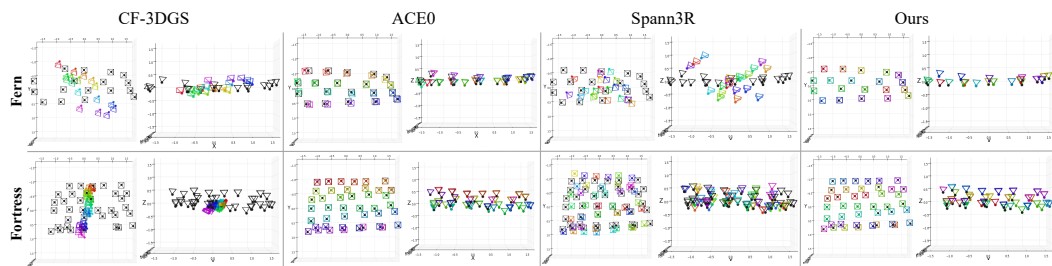

Figure 3: **Visualization of camera poses accuracy** on the LLFF dataset (Zoom in for best view). Black: pseudo-ground-truth camera poses obtained from COLMAP [47]. Colored: predicted camera poses.

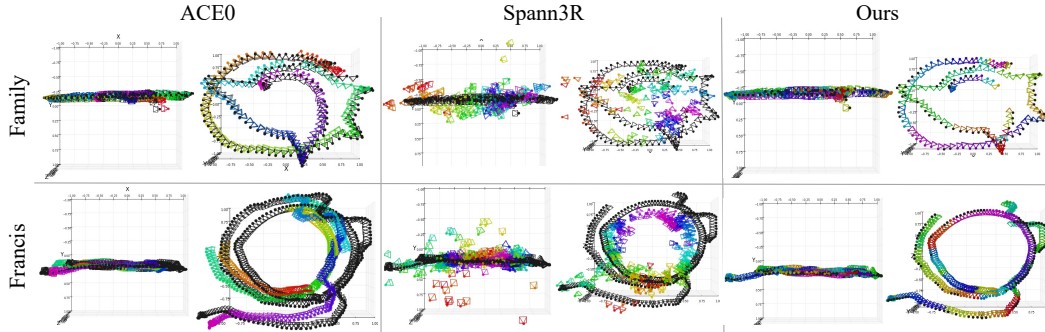

Figure 4: **Visualization of camera poses accuracy** on the Tanks-and-Temples dataset (Zoom in for best view). Black: pseudo-ground-truth camera poses obtained from COLMAP [47]. Colored: predicted camera poses.

**Evaluation Datasets.** We evaluate our method on the LLFF dataset [41], the Mip-NeRF360 dataset [3], the Tanks-and-Temples dataset [29], the sequential Tanks-and-Temples dataset (Seq TnT) [5], and the 7scenes dataset [48]. The LLFF dataset contains 8 scenes with cameras facing forward. The Mip-NeRF360 dataset contains different scenes where cameras are distributed evenly in 360 degrees in the 3D space. The Tanks-and-Temples dataset is similar to the Mip-NeRF360 dataset in camera poses distribution and scene scales, but with more illumination and appearance changes. Note that the Tanks-and-Temples dataset is different from the sequential Tanks-and-Temples dataset preprocessed by Nope-NeRF, where the Seq TnT dataset is extracted from video segments and contains fewer regions than the 360-degree Tanks-and-Temples dataset and thus requires fewer 3D Gaussians to represent and is less challenging. The 7Scenes dataset contains indoor scenes scanned with the Kinect camera.

Table 2: Camera pose accuracy averaged on each dataset. The red, orange and yellow colors respectively denote the best, the second best, and the third best results.

| Scenes | LLFF | | MipNeRF360 | | Tanks-and-Temples | | Seq TnT | |
|---|---|---|---|---|---|---|---|---|
| | $\Delta R$ | $\Delta t$ | $\Delta R$ | $\Delta t$ | $\Delta R$ | $\Delta t$ | $\Delta R$ | $\Delta t$ |
| NeRFmm [64] | 1.04 | 0.423 | 77.14 | 2.837 | 61.40 | 2.083 | 0.48 | 1.735 |
| BARF [32] | 0.45 | 0.331 | 76.38 | 3.371 | 69.50 | 1.227 | 0.44 | 1.046 |
| Nope-NeRF [5] | 0.45 | 0.184 | 77.12 | 1.659 | 58.33 | 1.078 | 0.04 | 0.080 |
| CF-3DGS [19] | 1.27 | 5.087 | – | – | – | – | 0.07 | 0.041 |
| ACE0 [9] | 3.20 | 0.043 | 0.83 | 0.014 | 9.13 | 0.170 | 9.52 | 0.103 |
| Spann3R [58] | 20.00 | 0.485 | 11.80 | 0.222 | 14.93 | 0.335 | 13.34 | 0.592 |
| Ours | 0.17 | 0.005 | 0.04 | 0.04 | 0.04 | 0.005 | 0.31 | 0.002 |

**Implementation Details.** We initialize the scene regressor network using the pre-trained Spann3R model [58]. During training, we use the image resolution of $512 \times 512$ for all the datasets. We use

a learning rate of $1e-5$ to finetune the scene regressor. We use $s_{\text{sim}} = 0.3$ to reject edges when building the similarity graph. We use DSAC [7, 8] to compute the camera poses for candidate image registration. Since DSAC supports only a single focal length, we modify it to use different focal lengths for the image x-axis and y-axis. We set the threshold of inlier number $s_{\text{inlier}}$ to $5,000$ and the threshold of reprojection error to be within 6 pixels in DSAC. For the seed initialization, we finetune the scene regressor by 500 iterations. During the incremental training, we finetune the scene regressor by 1,000 iterations on the LLFF dataset and 1,500 iterations on the Mip-NeRF360 dataset. For the novel view synthesis task, images are downsampled by 4 during training and inference. We use the CUDA rasterizer provided by [39] to enable the gradient computation of camera poses during rasterization.

**Camera Pose Accuracy.** To evaluate the camera pose accuracy, we use COLMAP poses as ground-truth and compare our method with NeRFmm [64], BARF [32], and Nope-NeRF [5], which are pose-free NeRF methods. We also compare our method to the pose-free 3DGS methods CF-3DGS [19]. Besides pose-free NeRF/3DGS-based methods, we also compare with the neural 3D reconstruction methods ACE0 [9] and Spann3R [58] with their official code. We present the quantitative results in Table 2. The unit for rotation error is degree, and the unit for translation error is dimensionless since the absolute scale in ground-truth is unknown. The results show that the camera pose accuracy of our method consistently outperforms all other methods. The pose-free NeRF methods perform very well on the LLFF dataset and the Seq TnT dataset, while failing on the MipNeRF360 dataset and the Tanks-and-Temples dataset. While CF-3DGS claims to be COLMAP-free, it performed badly on the LLFF dataset (*cf.* Fig. 3) and has similar performance on the Seq TnT dataset since it is only designed for handling sequential data. CF-3DGS also failed on the MipNeRF360 and Tanks-and-Temples dataset due to out-of-memory (marked by '-'). ACE0 achieved fairly good results on almost all the datasets, but the accuracy on the LLFF dataset and the Seq TnT dataset is not as good as other methods except Spann3R. We conjecture this is because these two datasets have only forward-facing cameras, which provide fewer multi-view constraints than the MipNeRF360 dataset. Notably, we found ACE0 has a large reconstruction error on the 'Francis' scene of the Tanks-and-Temples dataset (*cf.* the bottom row in Fig. 4), which suggests there is still room to improve ACE0 on robustness.

Table 3: Averaged Novel view synthesis results. COLMAP$^\star$ and ACE0$^\star$ denote training 3DGS with pose optimization. The red , orange , and yellow colors respectively denote the best, the second best, and the third best results.

| Scenes | LLFF | | | MipNeRF360 | | | Tanks-and-Temples | | | Seq TnT | | |
|---|---|---|---|---|---|---|---|---|---|---|---|---|
| | PSNR ↑ | SSIM ↑ | LPIPS ↓ | PSNR ↑ | SSIM ↑ | LPIPS ↓ | PSNR ↑ | SSIM ↑ | LPIPS ↓ | PSNR ↑ | SSIM ↑ | LPIPS ↓ |
| NeRFmm [64] | 22.84 | 0.640 | 0.460 | 11.23 | 0.410 | 0.812 | 10.31 | 0.332 | 0.881 | 22.50 | 0.59 | 0.54 |
| BARF [32] | 23.97 | 0.626 | 0.238 | 12.92 | 0.428 | 0.801 | 11.55 | 0.366 | 0.840 | 23.42 | 0.610 | 0.540 |
| Nope-NeRF [5] | 25.09 | 0.750 | 0.330 | 13.62 | 0.451 | 0.779 | 11.93 | 0.373 | 0.835 | 26.34 | 0.74 | 0.39 |
| CF-3DGS [19] | 16.52 | 0.479 | 0.437 | – | – | – | – | – | – | 31.28 | 0.93 | 0.09 |
| COLMAP [47] | 24.64 | 0.794 | 0.132 | 28.08 | 0.845 | 0.121 | 23.06 | 0.743 | 0.200 | 30.64 | 0.92 | 0.07 |
| COLMAP$^\star$ | 24.77 | 0.796 | 0.124 | 28.11 | 0.843 | 0.123 | 23.17 | 0.743 | 0.197 | 30.59 | 0.93 | 0.07 |
| ACE0 [9] | 22.74 | 0.709 | 0.182 | 24.31 | 0.663 | 0.269 | 20.93 | 0.661 | 0.302 | 24.32 | 0.81 | 0.19 |
| ACE0$^\star$ | 22.71 | 0.709 | 0.183 | 24.32 | 0.664 | 0.271 | 20.67 | 0.644 | 0.324 | 24.49 | 0.82 | 0.18 |
| Ours | 25.01 | 0.797 | 0.122 | 28.19 | 0.862 | 0.095 | 23.17 | 0.745 | 0.197 | 30.96 | 0.93 | 0.06 |

We also provide the recovered camera poses in Fig. 5 to show that our method also works on videos (left) and images with unevenly distributed camera poses (right).

**Novel View Synthesis.** We further evaluate our method on the task of novel view synthesis, which could provide better metrics to validate the accuracy of camera poses since COLMAP poses can contain errors. We provide the metrics of PSNR, SSIM [63], and LPIPS in Table 3. For COLMAP and ACE0, we use the same framework to train 3DGS. The pose-free NeRF methods have low numbers in the MipNeRF360 and Tanks-and-Temples dataset since they failed to estimate reliable camera poses. Surprisingly, we found our method obtains the best results in image rendering quality on almost all datasets and even outperforms COLMAP, showing that our method can compute more accurate camera poses than COLMAP. Though our method is not specifically designed for sequential images (in contrast to CF-3DGS), it still obtains comparable results to CF-3DGS and is even better than CF-3DGS in LPIPS. To further highlight the importance of our method, we conducted experiments to show the difficulty of jointly optimizing camera poses with 3DGS. We enable the gradients backpropagation of camera poses during rasterization when training 3DGS with COLMAP and ACE0 (denoted as COLMAP$^\star$ and ACE0$^\star$ respectively). From Tab. 3, we observe COLMAP$^\star$ is

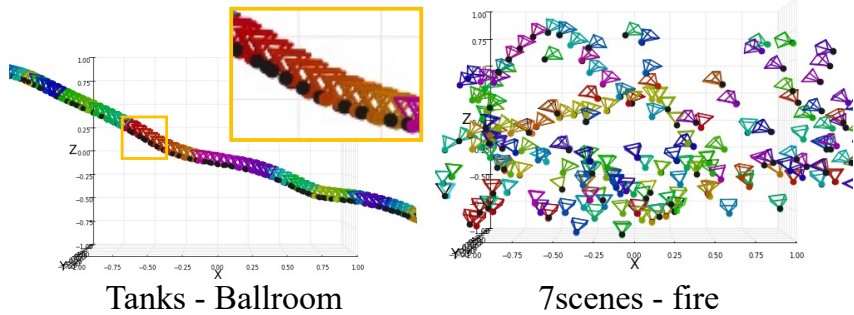

Tanks - Ballroom                7scenes - fire

Figure 5: Reconstructed camera poses of DeepGfM on the indoor 7scenes and sequential tanks-and-temples (Seq TnT) dataset.

slightly better than COLMAP – given highly accurate initial poses; In contrast, ACE0$^\star$ is even worse than ACE0 when initial poses are inaccurate. We also provide the qualitative results of novel view synthesis in Fig. 6, Fig. 7 and Fig. 8. More qualitative results are provided in our supplementary.

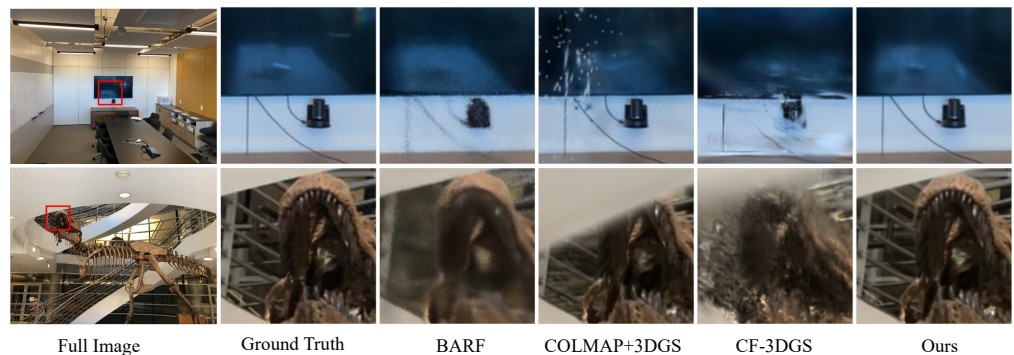

Full Image          Ground Truth          BARF          COLMAP+3DGS          CF-3DGS          Ours

Figure 6: The qualitative results of **novel view synthesis on LLFF forward-facing dataset [41]**.

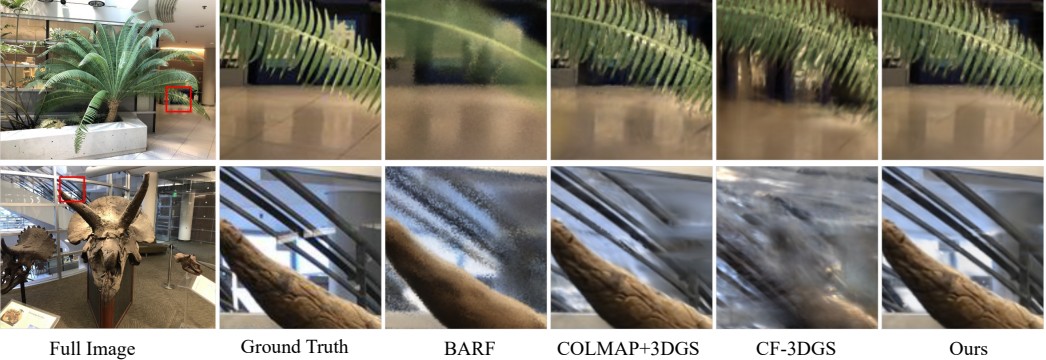

Full Image          Ground Truth          BARF          COLMAP+3DGS          CF-3DGS          Ours

Figure 7: **The qualitative results of novel view synthesis on LLFF forward-facing dataset [41]**.

**Ablation Study.** We ablate the effectiveness of our incremental training step and camera pose refinement step in Table 4, which are averaged on eight scenes of the LLFF dataset. We can see that with the refinement step, the camera pose accuracy is improved significantly. Although our proposed method is initialized from the pre-trained Spann3R model, the convergence of the refinement step can still fail in cases where the derived camera poses are grossly erroneous. We observe that the camera pose accuracy is improved with our incremental training pipeline (Ours$_{coarse}$ *v.s.* Spann3R), which provides much better initial values for refinement than simply from Spann3R.

We further ablate the effectiveness of finalizing the camera poses in our training pipeline in Table 5. The unit for rotation error is degree. We denote our method without the finalizing step as Ours$_{nf}$. We

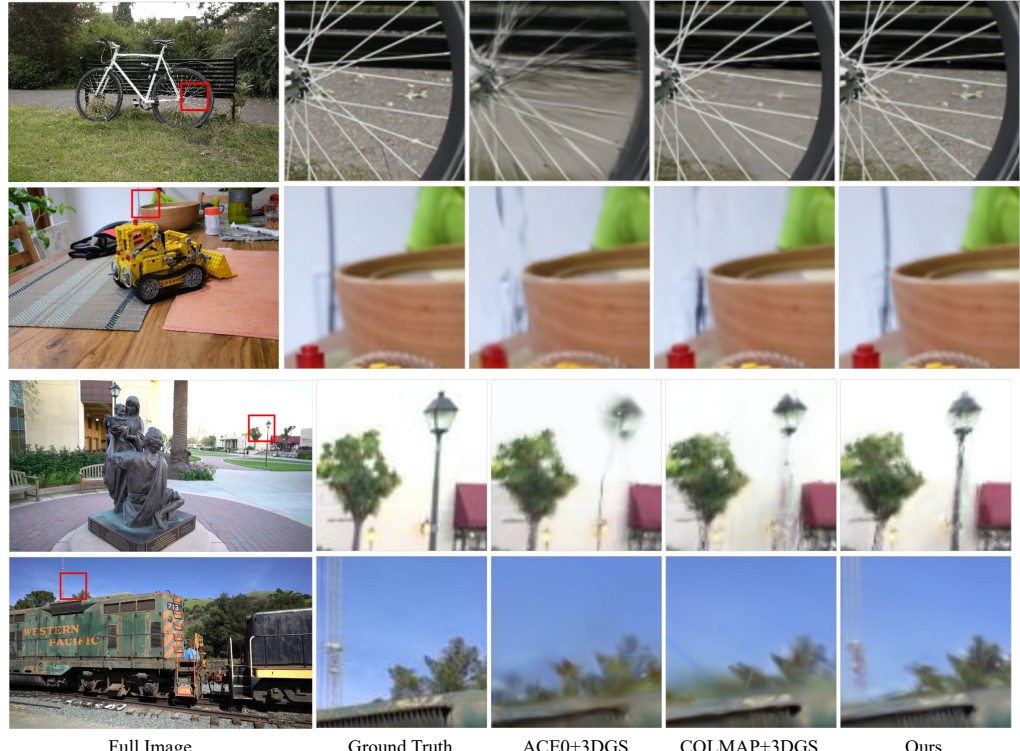

|  | Full Image | Ground Truth | ACE0+3DGS | COLMAP+3DGS | Ours |

Figure 8: The qualitative results of **novel view synthesis on the MipNeRF360 dataset** (top two rows) and **the Tanks-and-Temples dataset** (bottom two rows).

Table 4: Ablation study of **the camera pose refinement step**.

|  | Spann3R [58] | | Ours$_{coarse}$ | | Ours$_{refine}$ | |
|---|---|---|---|---|---|---|
|  | $\Delta R$ | $\Delta t$ | $\Delta R$ | $\Delta t$ | $\Delta R$ | $\Delta t$ |
| Avg | 20.00 | 0.485 | 10.20 | 0.229 | 0.17 | 0.005 |

Table 5: Ablation study of **the reconstruction finalization step**.

| Scenes | Bicycle | | Counter | | Garden | | Kitchen | |
|---|---|---|---|---|---|---|---|---|
|  | $\Delta R$ | $\Delta t$ | $\Delta R$ | $\Delta t$ | $\Delta R$ | $\Delta t$ | $\Delta R$ | $\Delta t$ |
| Ours$_{nf}$ | 0.096 | 0.015 | 0.041 | 0.007 | 0.095 | 0.012 | 0.150 | 0.219 |
| Ours | **0.035** | **0.005** | **0.029** | **0.002** | **0.028** | **0.002** | **0.052** | **0.008** |

can see that the finalization effectively mitigates the error accumulation in both camera rotations and translations. We also emphasize that the camera pose finalizing step is important to the neural scene refinement step since 3DGS is sensitive to even small perturbations in camera poses. Moreover, jointly optimizing explicit 3DGS and camera poses during training has limited effect when camera poses are close to ground truth and can even diverge the training [70].

## 5 Conclusion

In this paper, we propose DeepGfM to reconstruct neural scenes from unposed images. Our method adopts a pre-trained 3D foundation model as a scene regressor and leverages its learned geometry priors to ease the task of pose-free 3DGS training. Based on the learned geometry, we obtain coarse camera poses by RANSAC and PnP solver and refine them with a point-to-camera ray consistency loss. Our training pipeline incrementally registers the image batch into a training buffer and progressively finetunes the model in a self-supervised manner. Our method surpassed state-of-the-art pose-free NeRF/3DGS methods and even outperforms COLMAP-based 3DGS on multiple datasets.

**Limitations.** Though our method can produce higher quality reconstruction results in both camera poses and novel view synthesis, it requires more GPU memory and training time than ACE0 since we are fine-tuning transformers. In addition, our method uses Spann3R as the backbone of the scene regressor. Therefore, the training speed and the convergence rate depend on the performance of Spann3R. Moreover, when reconstructing larger-scale scenes, the spatial memory bank would require more GPU memory, which limits its application to larger scenes. To solve these limitations, future work may include the use of better variants of DUSt3R (*e.g.*, Fast3R [68] and CUT3R [61]) to enable reconstruction on larger scenes, adopting LORA [22, 38] to improve fine-tuning efficiency.

**Acknowledgement.** This research / project is supported by the National Research Foundation (NRF) Singapore, under its NRF-Investigatorship Programme (Award ID. NRF-NRFI09-0008). Yu Chen is also partially supported by a Google PhD Fellowship.

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

# A Implementation

**Neural Scene Regressor.** We initialize part of our model with the pretrained weights from Spann3R [58]. Following Spann3R, we use a ViT-large encoder, two ViT-base decoders, and a DPT head [44] for predicting dense pointmaps. We additionally use another DPT head to predict 3D Gaussian primitives. Though Spann3R is trained on images with resolution $224 \times 224$, we finetune it on the reconstructed scene with image resolution $512 \times 512$ using AdamW [28] optimizer.

**Image Registration.** We use DSAC to register images and compute coarse camera poses. An image is successfully registered if it has at least $5,000$ inliers with a reprojection threshold of 6 px. We use 64 hypotheses and an inlier alpha of $100$. To accelerate registration, the dense pointmaps for each image are downsampled by $4$. To further speed up the training and reduce memory footprint, we do not use all pointmaps from the newly registered images to refine camera poses. Instead, we pre-build sparse point tracks $\mathbf{X}_k = \{(\mathbf{y}_{ij}, \mathbf{I}_i)\}$, where $\mathbf{y}_{ij}$ denotes the $j$-th pixels observed on image $\mathbf{I}_i$. We adopt the Union-Find algorithm to remove duplicate and ambiguous tracks to improve the robustness during refinement. We adopt the Huber loss with a threshold of $0.1$ as the robust loss function in Eq. (8).

**Pseudo Algorithm of Our Training Pipeline.** We provide the pseudo algorithm of our incremental training pipeline as described in Sec. 3.3 in Alg. 1. At line 1, $\mathcal{V}_i$ denotes the set of graph nodes, and $\mathcal{E}_i$ denotes the set of graph edges. At line 4, $|\cdot|$ denotes the capacity of a set. We align our predicted camera poses to pseudo-ground-truth using the Umeyama [56] algorithm. Note that the camera poses and sparse points of COLMAP are normalized at the end of reconstruction. In line 14, we also normalize our predicted camera poses and dense points before refining the neural scene for fair comparison. We experimentally found that this can improve the training stability of 3DGS.

---

**Algorithm 1** Incremental Neural Reconstruction Algorithm

---

**Require:** a set of (unordered) images $\{\mathbf{I}_i\}$, maximum iteration per epoch $\text{iter}_{\max}$
**Ensure:** Camera poses $\{\mathbf{T}_i\}$, 3D Gaussian primitives $\{\mathcal{G}_i\}$
  1: Construct a similarity graph $\mathcal{G}_{\text{sim}} = (\mathcal{V}_i, \mathcal{E}_i)$
  2: Initialization from a seed image $\mathbf{I}_{\text{seed}}$ (*cf.* Sec. 3.3.1)
  3: Registered image set $(\mathcal{I}_{\text{reg}}, \mathcal{T}_{\text{reg}}) = \{(\mathbf{I}_{\text{seed}}, \mathbf{T}_{\text{seed}})\}$
  4: **while** $|\mathcal{I}_{\text{reg}}| < |\{\mathbf{I}_i\}|$ **do**
  5:      Register a new image batch $(\mathcal{I}_{\text{new}}, \mathcal{T}_{\text{new}})$ by Eq. (7)
  6:      Refine newly registered camera poses by Eq. (8)
  7:      Update training buffer using $(\mathcal{I}_{\text{new}}, \mathcal{T}_{\text{new}})$
  8:      $\text{iter} := 0$
  9:      **while** $\text{iter} < \text{iter}_{\max}$ **do**
10:         Finetune scene regressor $f_{\text{SCR}}$ using Eq. (9)
11:         $\text{iter} := \text{iter} + 1$
12:      $\mathcal{I}_{\text{reg}} := \mathcal{I}_{\text{reg}} + \mathcal{I}_{\text{new}}, \ \mathcal{T}_{\text{reg}} := \mathcal{T}_{\text{reg}} + \mathcal{T}_{\text{new}}$
13: Finalize camera poses $\mathcal{T}_{\text{reg}} = \{\mathbf{T}_i\}$ using (8)
14: Normalize camera poses $\mathcal{T}_{\text{reg}}^{\text{norm}} = \text{Normalize}(\mathcal{T}_{\text{reg}})$
15: Finalize neural scene $\{\mathcal{G}_i\}$ (*cf.* Sec. 3.3.3)

---

# B Additional Qualitative Results

We present more qualitative results in this Section. More reconstruction results of camera pose and pointmaps are included in Fig. 17.

**Ablation of Pose Refinement.** We present the visual comparison of our method with (Ours$_{\text{refine}}$) and without (Ours$_{\text{coarse}}$) the refinement step in Fig. 9. As shown in Fig. 9, camera poses are aligned closer to the ground truth after camera pose refinement. Compared to the camera poses obtained from Spann3R of the first row in Fig. 3 and the third row in Fig. 10, the coarse camera poses are closer to the ground truth, which is coherent with the quantitative results provided in Table 6.

Table 6: **Ablation study of camera pose accuracy**.

| Scenes | Spann3R [58] | | Ours_coarse | | Ours_refine | |
|---|---|---|---|---|---|---|
| | $\Delta R$ | $\Delta t$ | $\Delta R$ | $\Delta t$ | $\Delta R$ | $\Delta t$ |
| Fern | 39.03 | 0.767 | 01.30 | 0.125 | 0.26 | 0.005 |
| Flower | 11.91 | 0.285 | 16.53 | 0.609 | 0.52 | 0.011 |
| Fortress | 08.31 | 0.152 | 06.07 | 0.127 | 0.04 | 0.002 |
| Horns | 06.98 | 0.349 | 14.23 | 0.145 | 0.03 | 0.001 |
| Leaves | 44.09 | 0.801 | 18.24 | 0.187 | 0.22 | 0.006 |
| Orchids | 09.77 | 0.256 | 07.22 | 0.255 | 0.24 | 0.006 |
| Room | 07.48 | 0.513 | 10.22 | 0.180 | 0.03 | 0.001 |
| Trex | 32.39 | 0.758 | 07.76 | 0.210 | 0.03 | 0.010 |
| Avg | 20.00 | 0.485 | 10.20 | 0.229 | 0.17 | 0.005 |

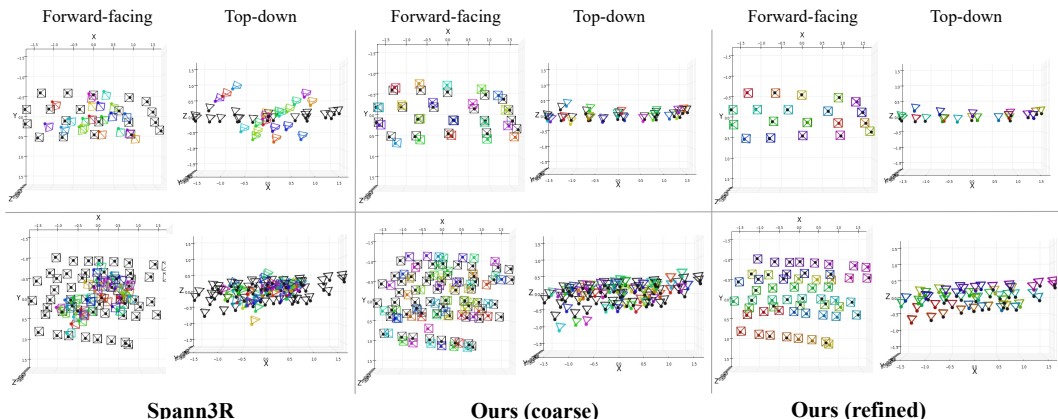

Figure 9: **Ablation of camera poses refinement on the LLFF dataset** (Zoom in for best view). Top and bottom are respectively the camera poses of the 'Fern' and 'Room' scenes.

**More Results of Camera Poses.** We provide more visual comparison of camera poses on the LLFF dataset in Fig. 10. The results on the MipNeRF360 dataset and more results on the Tanks-and-Temples dataset are respectively provided in Fig. 11 and Fig. 12. We can observe that CF-3DGS [19] failed to produce faithfully camera poses on the LLFF dataset, which has been analyzed in the main paper. While ACE0 [9] performs very well on the MipNeRF360 dataset, it struggles on the challenging Tanks-and-Temples dataset (*cf.* The 'Train' scene in Fig. 12). Moreover, we find that ACE0 performs poorly on the LLFF dataset. This may be due to the MLP decoder of ACE0 maps individual pixels to 3D space, while it requires well-distributed training views to constrain the network. However, the LLFF dataset contains only forward-facing cameras, which do not provide strong constraints from different view directions and therefore degenerate the training of ACE0.

We further provide the camera poses of our method on the indoor 7scenes dataset in Fig. 13, and the camera poses of our method on the sequential tanks-and-temples dataset in Fig. 14 to show that our method can work on diverse scenes and video sequences.

Table 7: **Quantitative results of camera pose accuracy on MipNeRF360 dataset**.

| Scenes | ACE0 [9] | | Spann3R [58] | | Ours | |
|---|---|---|---|---|---|---|
| | $\Delta R$ | $\Delta t$ | $\Delta R$ | $\Delta t$ | $\Delta R$ | $\Delta t$ |
| Bicycle | 1.02 | 0.017 | 10.79 | 0.212 | 0.035 | 0.005 |
| Counter | 0.83 | 0.014 | 11.16 | 0.226 | 0.029 | 0.002 |
| Garden | 0.82 | 0.012 | 15.65 | 0.279 | 0.028 | 0.002 |
| Kitchen | 0.64 | 0.012 | 9.60 | 0.171 | 0.052 | 0.008 |
| Avg | 0.83 | 0.014 | 11.8 | 0.222 | 0.04 | 0.004 |

**More Qualitative Results of Novel View Synthesis.** We present more qualitative results on the novel view synthesis task in Fig. 15, and Fig. 16. The quantitative camera pose accuracy only reveals how close the predicted camera poses are to the COLMAP poses. However, it cannot distinguish which is more accurate since COLMAP poses are only pseudo-ground-truth and it can produce

Table 8: **Quantitative results of camera pose accuracy on Tanks-and-Temples dataset**.

| Scenes | ACE0 | | Spann3R | | Ours | |
|---|---|---|---|---|---|---|
| | $\Delta R$ | $\Delta t$ | $\Delta R$ | $\Delta t$ | $\Delta R$ | $\Delta t$ |
| Family | 6.41 | 0.088 | 16.98 | 0.378 | 0.036 | 0.003 |
| Francis | 21.63 | 0.351 | 14.19 | 0.361 | 0.030 | 0.002 |
| Ignatius | 2.96 | 0.043 | 11.23 | 0.313 | 0.028 | 0.002 |
| Train | 5.52 | 0.196 | 17.33 | 0.286 | 0.065 | 0.011 |
| Avg | 9.13 | 0.170 | 14.93 | 0.335 | 0.04 | 0.005 |

Table 9: **Quantitative results of camera pose accuracy on 7scenes dataset**.

| Scenes | ACE0 | | Spann3R | | Ours | |
|---|---|---|---|---|---|---|
| | $\Delta R$ | $\Delta t$ | $\Delta R$ | $\Delta t$ | $\Delta R$ | $\Delta t$ |
| chess | 0.590 | 0.016 | 6.468 | 0.300 | 0.082 | 0.010 |
| fire | 0.680 | 0.016 | 8.662 | 0.346 | 0.116 | 0.004 |
| heads | 1.141 | 0.038 | 9.258 | 0.464 | 0.420 | 0.012 |
| pumpkin | 0.616 | 0.022 | 7.731 | 0.551 | 0.141 | 0.012 |
| Avg | 0.757 | 0.023 | 8.033 | 0.415 | 0.440 | 0.010 |

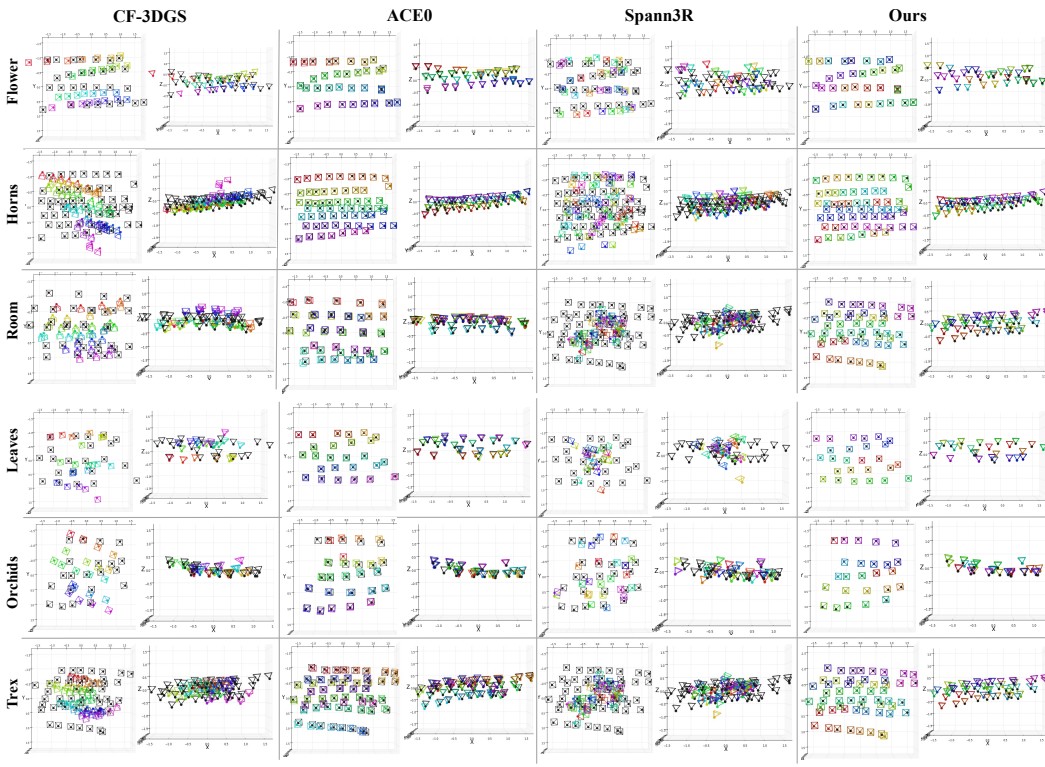

Figure 10: More **qualitative comparisons of camera poses accuracy on the LLFF dataset** (Zoom in for best view). Black: pseudo-ground-truth camera poses obtained from COLMAP [47]. Colored: predicted camera poses.

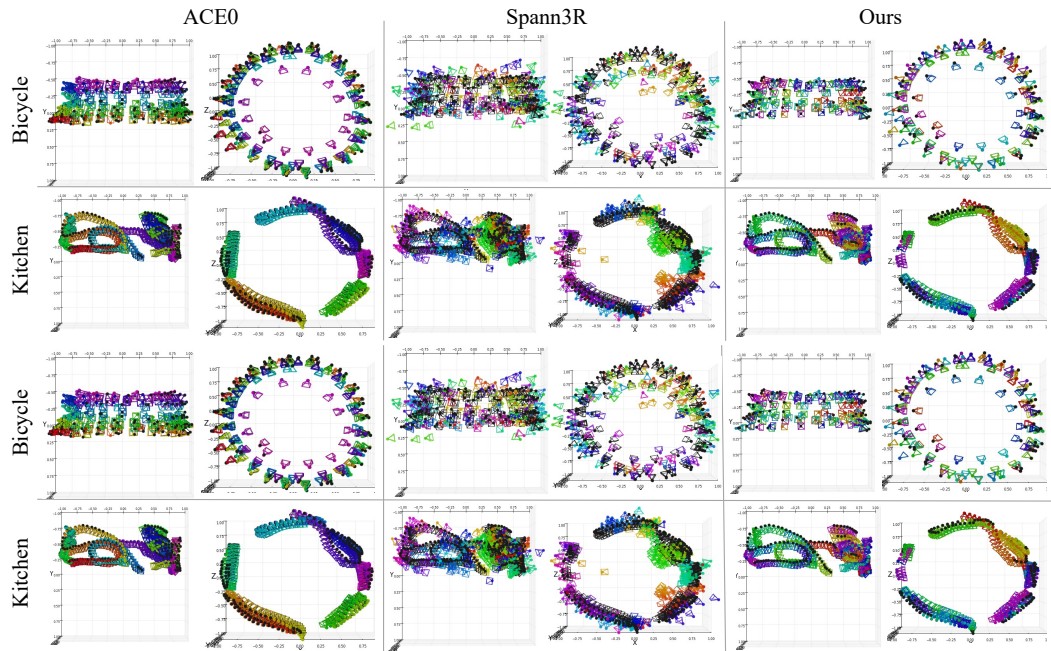

Figure 11: More **qualitative comparisons of camera poses accuracy on the MipNeRF360 dataset** (Zoom in for best view). Black: pseudo-ground-truth camera poses obtained from COLMAP [47]. Colored: predicted camera poses.

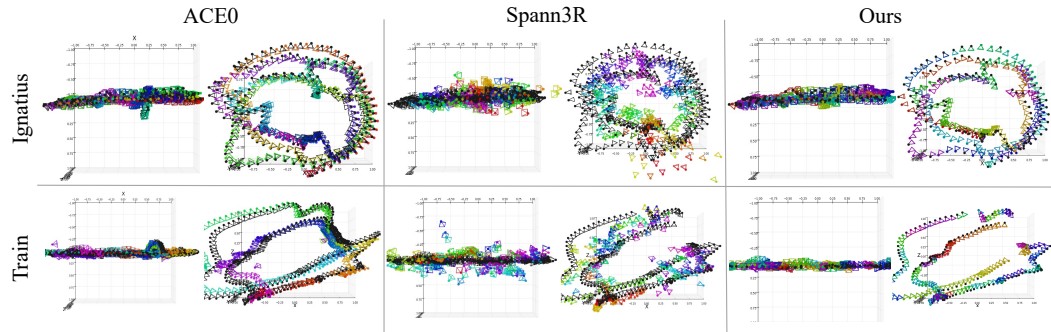

Figure 12: More **qualitative comparisons of camera poses accuracy on the Tanks-and-Temples dataset** (Zoom in for best view). Black: pseudo-ground-truth camera poses obtained from COLMAP [47]. Colored: predicted camera poses.

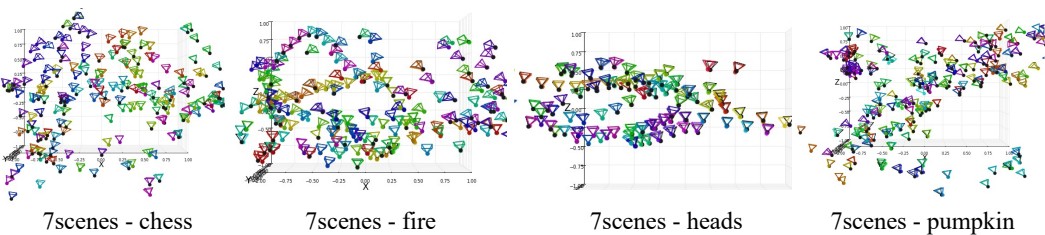

Figure 13: Visualizations of our camera poses on the 7scenes dataset.

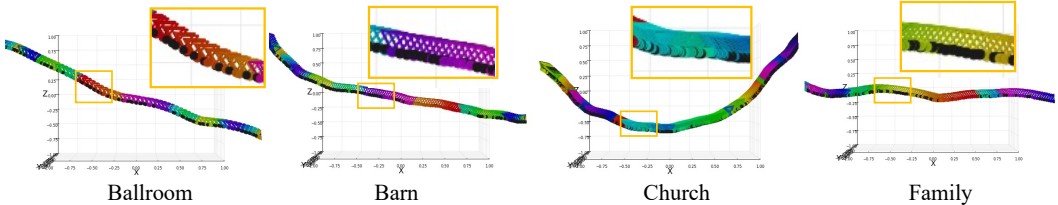

| Ballroom | Barn | Church | Family |

Figure 14: Visualizations of our camera poses on the Sequential Tanks-and-Temples dataset.

erroneous camera poses. Nonetheless, the results of novel view synthesis provide better metrics to show which one is better when two camera poses are close. In Fig. 15 and Fig. 16, we can observe that our method can render finer details when we zoom into the same areas. The visual comparison also provides coherent support to the quantitative results of novel view synthesis in Table 15 and Table 16.

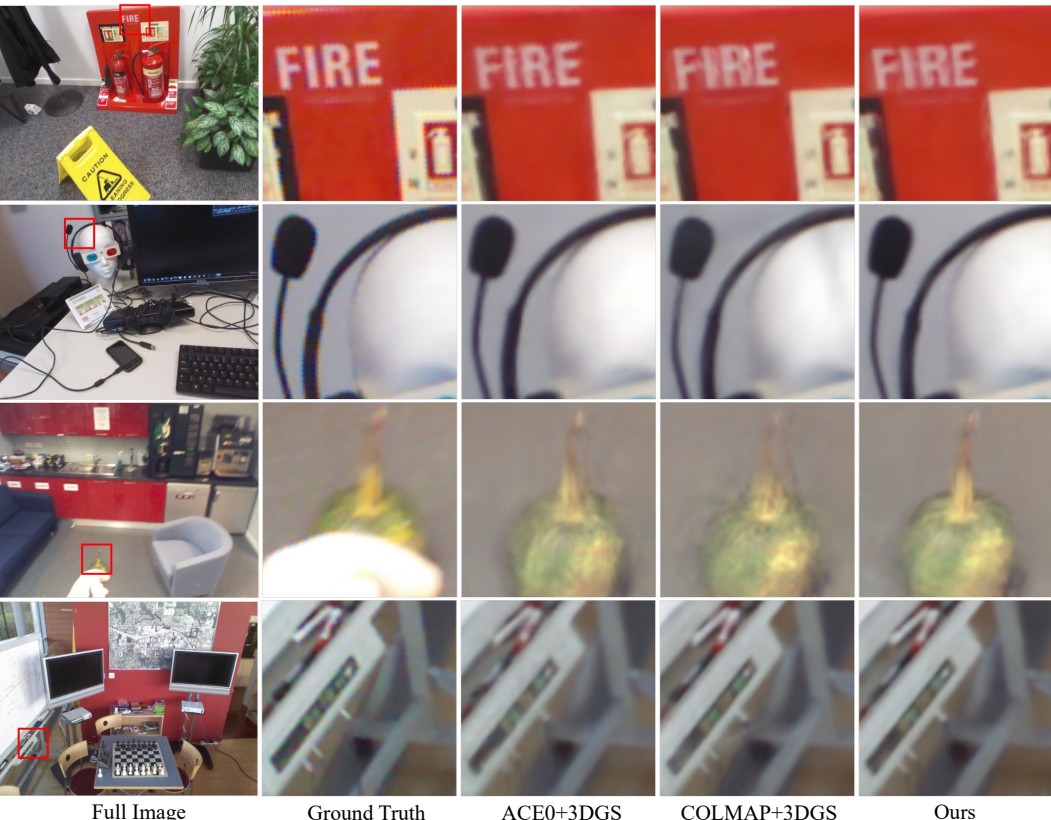

| Full Image | Ground Truth | ACE0+3DGS | COLMAP+3DGS | Ours |

Figure 15: More **qualitative comparisons of novel view synthesis on the 7Scenes dataset**. From top to bottom are scenes of fire, heads, pumpkin, and chess.

## C   More Discussion

**Differences with (neural) incremental SfM methods.**   Our incremental training pipeline is similar to the classical incremental SfM method [47] but differs as follows:

- *Seed Initialization*. Incremental SfM initializes from an image pair, where the camera poses of the image pair are fixed after initialization to fix the gauge freedom. However, our method initializes from a pretrained model and only one seed image.

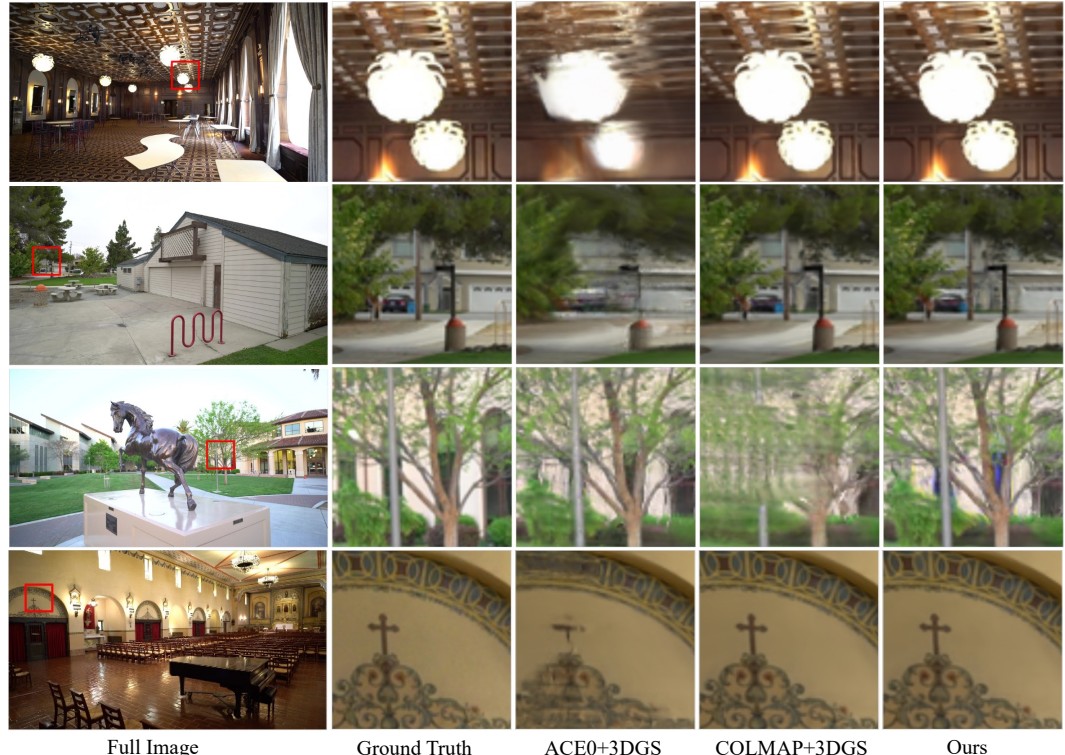

| Full Image | Ground Truth | ACE0+3DGS | COLMAP+3DGS | Ours |

Figure 16: More **qualitative comparisons of novel view synthesis on the sequential Tanks-and-Temples dataset**. From top to bottom are the scenes of the ballroom, barn, horse, and church.

- *Objective Function.* We fine-tune our model using a rendering loss while scenes are optimized by the reprojection error in SfM.

- *Scene Sparsity.* Our model predicts dense scene geometries, while SfM outputs sparse scene structures.

Our incremental training pipeline also shares some similarities with a recent learning-based SfM method ACE0 [9], with several key differences: 1) ACE0 only predicts sparse pointmaps, while our method predicts dense pointmaps and 3D Gaussian primitives. 2) ACE0 uses 2D CNN and MLP as the neural scene representation, while we use transformers as the scene representation. 3) The training batch of ACE0 is composed of pixels from multiple views, while our method takes as input image pairs in the training batch. Moreover, fine-tuning a pre-trained foundation model such as DUSt3R is not easy. This is because these 3D foundation models are supervised by ground-truth 3D points that are difficult to obtain in unseen scenes.

**Run Time and Memory Footprint.** Our pipeline converges faster than COLMAP [47]. On the LLFF, 7Scenes and Sequential Tanks-and-Temples, our method converges in two epochs, which takes about 25 minutes for each scene; On the MipNeRF360 dataset and the Tanks-and-Temples dataset, our method converges in $5 - 15$ epochs, which takes about 2 hours for each scene. We evaluate the model during training and save intermediate results to disk for every $1,000$ iteration. The evaluation time is also included in the training step. Our method takes about 21GB with a batch size of 1 during training on an NVIDIA 4090 GPU.

## D  Per-Scene Breakdown

We provide per-scene quantitative results for both camera pose accuracy in Table 10, Table 7, Table 8, Table 9, and Table 18, as well as novel view synthesis in Table 11, Table 14, Table 15, Table 16, Table 17, Table 19, and Table 20. Note that, for the MipNeRF360, Tanks-and-Temples, and 7Scenes

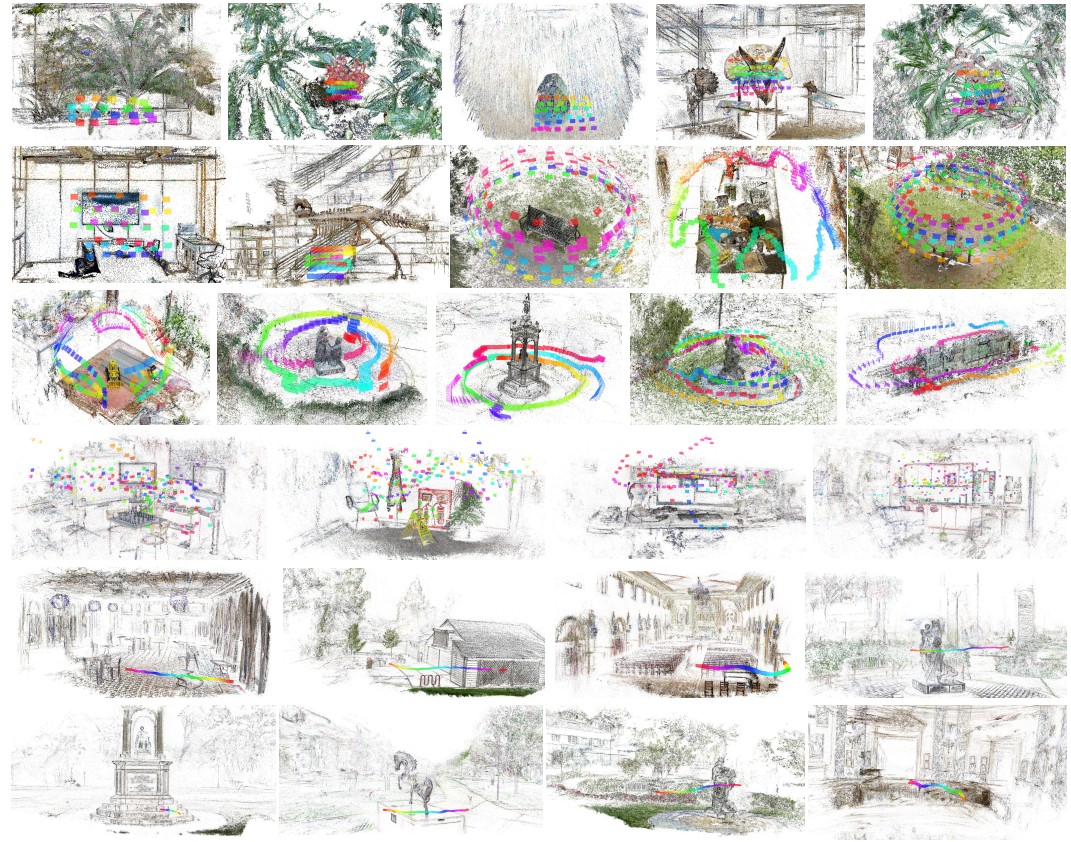

Figure 17: More **visual reconstruction results** on real-world datasets.

datasets, NeRFmm, BARF, and Nope-NeRF failed to obtain reasonable camera poses and clear novel views, thus, we did not present the per-scene quantitative results in tables for readability.

Table 10: **Quantitative results of camera pose accuracy on LLFF dataset**.

| Scenes | NeRFmm | | Nope-NeRF | | BARF [32] | | DBARF [12] | | ACE0 [9] | | CF-3DGS [19] | | Spann3R [58] | | Ours | |
|---|---|---|---|---|---|---|---|---|---|---|---|---|---|---|---|---|
| | $\Delta R$ | $\Delta t$ | $\Delta R$ | $\Delta t$ | $\Delta R$ | $\Delta t$ | $\Delta R$ | $\Delta t$ | $\Delta R$ | $\Delta t$ | $\Delta R$ | $\Delta t$ | $\Delta R$ | $\Delta t$ | $\Delta R$ | $\Delta t$ |
| Fern | 1.82 | 0.706 | 0.99 | 0.252 | 0.19 | 0.192 | 0.89 | 0.341 | 7.46 | 0.064 | 2.81 | 9.254 | 39.03 | 0.767 | 0.26 | 0.005 |
| Flower | 0.42 | 0.086 | 0.10 | 0.035 | 0.25 | 0.224 | 1.39 | 0.318 | 6.73 | 0.031 | 0.24 | 2.586 | 11.91 | 0.285 | 0.52 | 0.011 |
| Fortress | 0.74 | 0.233 | 0.30 | 0.081 | 0.48 | 0.364 | 0.59 | 0.229 | 0.98 | 0.010 | 1.28 | 8.592 | 8.31 | 0.152 | 0.04 | 0.002 |
| Horns | 0.85 | 0.321 | 0.45 | 0.217 | 0.30 | 0.222 | 0.82 | 0.292 | 1.48 | 0.013 | 1.15 | 2.371 | 6.98 | 0.349 | 0.03 | 0.001 |
| Leaves | 0.05 | 0.138 | 0.14 | 0.218 | 1.27 | 0.249 | 4.63 | 0.855 | 4.93 | 0.036 | 0.33 | 7.350 | 44.09 | 0.801 | 0.22 | 0.006 |
| Orchids | 2.03 | 0.686 | 0.38 | 0.203 | 0.63 | 0.404 | 1.16 | 0.573 | 1.19 | 0.093 | 1.45 | 2.772 | 9.77 | 0.256 | 0.24 | 0.006 |
| Room | 1.66 | 0.670 | 0.94 | 0.244 | 0.32 | 0.270 | 0.53 | 0.360 | 1.06 | 0.041 | 1.36 | 3.336 | 7.48 | 0.513 | 0.03 | 0.001 |
| Trex | 0.78 | 0.542 | 0.32 | 0.219 | 0.14 | 0.720 | 1.06 | 0.463 | 1.73 | 0.059 | 1.56 | 4.431 | 32.39 | 0.758 | 0.03 | 0.010 |
| Avg | 1.04 | 0.423 | 0.45 | 0.184 | 0.45 | 0.331 | 1.38 | 0.429 | 3.20 | 0.043 | 1.27 | 5.087 | 20.00 | 0.485 | 0.17 | 0.005 |

Table 11: **Quantitative results of novel view synthesis on LLFF dataset**.

| Scenes | NeRFmm | | | Nope-NeRF | | | BARF | | | CF-3DGS | | | 3DGS | | | ACE0 | | | Ours | | |
|---|---|---|---|---|---|---|---|---|---|---|---|---|---|---|---|---|---|---|---|---|---|
| | PSNR↑ | SSIM↑ | LPIPS↓ | PSNR↑ | SSIM↑ | LPIPS↓ | PSNR↑ | SSIM↑ | LPIPS↓ | PSNR↑ | SSIM↑ | LPIPS↓ | PSNR↑ | SSIM↑ | LPIPS↓ | PSNR↑ | SSIM↑ | LPIPS↓ | PSNR↑ | SSIM↑ | LPIPS↓ |
| Fern | 20.58 | 0.590 | 0.500 | 23.01 | 0.710 | 0.380 | 23.79 | 0.710 | 0.311 | 17.35 | 0.494 | 0.428 | 23.63 | 0.794 | 0.136 | 19.33 | 0.585 | 0.295 | 23.77 | 0.797 | 0.114 |
| Flower | 27.02 | 0.760 | 0.320 | 29.39 | 0.860 | 0.190 | 23.37 | 0.698 | 0.211 | 20.17 | 0.622 | 0.362 | 26.91 | 0.829 | 0.096 | 25.84 | 0.787 | 0.109 | 25.81 | 0.812 | 0.108 |
| Fortress | 24.94 | 0.570 | 0.570 | 29.38 | 0.800 | 0.280 | 29.08 | 0.823 | 0.132 | 14.73 | 0.395 | 0.460 | 29.93 | 0.880 | 0.078 | 29.09 | 0.848 | 0.085 | 29.31 | 0.868 | 0.084 |
| Horns | 23.67 | 0.660 | 0.480 | 25.24 | 0.730 | 0.370 | 22.78 | 0.727 | 0.298 | 15.60 | 0.412 | 0.514 | 26.02 | 0.862 | 0.121 | 24.99 | 0.821 | 0.137 | 26.67 | 0.882 | 0.093 |
| Leaves | 19.46 | 0.550 | 0.460 | 19.85 | 0.600 | 0.400 | 18.78 | 0.537 | 0.353 | 15.38 | 0.416 | 0.398 | 18.98 | 0.593 | 0.205 | 16.15 | 0.419 | 0.314 | 19.55 | 0.640 | 0.147 |
| Orchids | 16.77 | 0.400 | 0.550 | 19.51 | 0.560 | 0.430 | 19.45 | 0.574 | 0.291 | 13.80 | 0.258 | 0.516 | 18.98 | 0.612 | 0.159 | 17.09 | 0.527 | 0.242 | 16.45 | 0.526 | 0.270 |
| Room | 26.14 | 0.840 | 0.390 | 28.54 | 0.890 | 0.280 | 31.95 | 0.940 | 0.099 | 18.36 | 0.713 | 0.382 | 28.96 | 0.927 | 0.115 | 26.89 | 0.894 | 0.118 | 32.37 | 0.948 | 0.073 |
| Trex | 24.13 | 0.770 | 0.390 | 25.82 | 0.840 | 0.290 | 22.55 | 0.767 | 0.206 | 16.76 | 0.522 | 0.434 | 24.74 | 0.881 | 0.145 | 22.52 | 0.792 | 0.155 | 26.11 | 0.905 | 0.084 |
| Avg | 22.84 | 0.640 | 0.460 | 25.09 | 0.750 | 0.330 | 23.97 | 0.626 | 0.238 | 16.52 | 0.479 | 0.437 | 24.64 | 0.794 | 0.132 | 22.74 | 0.709 | 0.182 | 25.01 | 0.797 | 0.122 |

# E   Social Impacts

Since it is the era of large foundation models, it is popular for more and more works that dive into training a generalizable geometric foundation model and then use it to reconstruct scenes in a fast

Table 12: **Quantitative results of novel view synthesis on MipNeRF360 dataset**.

| Scenes | 3DGS | | | +exposure | | | +depth | | | +exposure+depth | | | +ae | | | +ae+exposure | | | +ae+exposure+depth | | |
|---|---|---|---|---|---|---|---|---|---|---|---|---|---|---|---|---|---|---|---|---|---|
| | PSNR↑ | SSIM↑ | LPIPS↓ | PSNR↑ | SSIM↑ | LPIPS↓ | PSNR↑ | SSIM↑ | LPIPS↓ | PSNR↑ | SSIM↑ | LPIPS↓ | PSNR↑ | SSIM↑ | LPIPS↓ | PSNR↑ | SSIM↑ | LPIPS↓ | PSNR↑ | SSIM↑ | LPIPS↓ |
| Bicycle | 24.93 | 0.731 | 0.186 | 25.01 | 0.733 | 0.184 | 24.88 | 0.729 | 0.188 | 24.96 | 0.731 | 0.185 | 25.34 | 0.735 | 0.205 | 25.34 | 0.733 | 0.203 | 25.14 | 0.728 | 0.211 |
| Bonsai | 31.38 | 0.938 | 0.060 | 31.54 | 0.938 | 0.059 | 31.34 | 0.937 | 0.061 | 31.50 | 0.938 | 0.060 | 34.42 | 0.959 | 0.052 | 34.91 | 0.961 | 0.049 | 34.38 | 0.958 | 0.055 |
| Counter | 29.46 | 0.916 | 0.087 | 29.43 | 0.916 | 0.088 | 29.12 | 0.912 | 0.093 | 29.20 | 0.912 | 0.092 | 30.39 | 0.923 | 0.083 | 30.47 | 0.924 | 0.081 | 30.11 | 0.919 | 0.088 |
| Flower | 21.66 | 0.595 | 0.352 | 21.64 | 0.596 | 0.354 | 21.68 | 0.595 | 0.349 | 21.64 | 0.593 | 0.353 | 21.48 | 0.587 | 0.357 | 21.49 | 0.585 | 0.359 | 21.25 | 0.572 | 0.364 |
| Garden | 27.15 | 0.850 | 0.084 | 27.23 | 0.851 | 0.084 | 27.21 | 0.851 | 0.083 | 27.15 | 0.850 | 0.085 | 27.68 | 0.858 | 0.089 | 27.91 | 0.862 | 0.085 | 27.76 | 0.859 | 0.089 |
| Kitchen | 30.40 | 0.920 | 0.047 | 30.30 | 0.920 | 0.047 | 29.70 | 0.914 | 0.051 | 29.88 | 0.914 | 0.051 | 33.19 | 0.955 | 0.035 | 33.45 | 0.956 | 0.034 | 31.85 | 0.947 | 0.041 |
| Room | 31.84 | 0.931 | 0.076 | 32.02 | 0.931 | 0.075 | 31.84 | 0.930 | 0.076 | 31.93 | 0.930 | 0.076 | 33.53 | 0.942 | 0.070 | 33.75 | 0.943 | 0.068 | 33.35 | 0.941 | 0.071 |
| Stump | 26.59 | 0.758 | 0.166 | 26.65 | 0.760 | 0.165 | 26.50 | 0.755 | 0.164 | 26.60 | 0.757 | 0.163 | 26.38 | 0.744 | 0.178 | 26.38 | 0.744 | 0.177 | 26.15 | 0.736 | 0.182 |
| Treehill | 23.00 | 0.620 | 0.352 | 23.19 | 0.625 | 0.338 | 22.75 | 0.613 | 0.356 | 22.88 | 0.618 | 0.349 | 22.86 | 0.614 | 0.359 | 23.03 | 0.618 | 0.357 | 23.03 | 0.618 | 0.357 |
| Avg | 27.38 | 0.807 | 0.157 | 27.45 | 0.808 | 0.155 | 27.22 | 0.804 | 0.158 | 27.30 | 0.805 | 0.157 | 28.36 | 0.813 | 0.159 | 28.53 | 0.814 | 0.157 | 28.11 | 0.809 | 0.162 |

Table 13: **Quantitative results of novel view synthesis on Tanks-and-Temples dataset**.

| Scenes | 3DGS | | | +exposure | | | +depth | | | +exposure+depth | | | +ae | | | +ae+exposure | | | +ae+exposure+depth | | |
|---|---|---|---|---|---|---|---|---|---|---|---|---|---|---|---|---|---|---|---|---|---|
| | PSNR↑ | SSIM↑ | LPIPS↓ | PSNR↑ | SSIM↑ | LPIPS↓ | PSNR↑ | SSIM↑ | LPIPS↓ | PSNR↑ | SSIM↑ | LPIPS↓ | PSNR↑ | SSIM↑ | LPIPS↓ | PSNR↑ | SSIM↑ | LPIPS↓ | PSNR↑ | SSIM↑ | LPIPS↓ |
| Barn | 25.79 | 0.809 | 0.191 | 25.87 | 0.809 | 0.191 | 25.38 | 0.797 | 0.205 | 25.58 | 0.798 | 0.203 | 26.65 | 0.821 | 0.184 | 26.78 | 0.821 | 0.182 | 26.38 | 0.809 | 0.194 |
| Caterpillar | 23.26 | 0.727 | 0.248 | 23.83 | 0.738 | 0.234 | 23.20 | 0.724 | 0.250 | 23.74 | 0.735 | 0.238 | 24.17 | 0.745 | 0.237 | 24.03 | 0.739 | 0.235 | 23.90 | 0.733 | 0.244 |
| Family | 18.85 | 0.684 | 0.267 | 19.52 | 0.697 | 0.253 | 21.09 | 0.741 | 0.195 | 21.46 | 0.748 | 0.190 | 19.71 | 0.681 | 0.247 | 20.00 | 0.684 | 0.242 | 22.45 | 0.745 | 0.183 |
| Francis | 24.98 | 0.830 | 0.226 | 25.60 | 0.834 | 0.222 | 26.13 | 0.845 | 0.200 | 27.00 | 0.854 | 0.188 | 26.76 | 0.847 | 0.204 | 26.19 | 0.838 | 0.216 | 27.60 | 0.853 | 0.190 |
| Horse | 22.29 | 0.826 | 0.162 | 22.61 | 0.828 | 0.155 | 22.53 | 0.825 | 0.155 | 22.66 | 0.826 | 0.152 | 22.59 | 0.828 | 0.159 | 23.60 | 0.835 | 0.145 | 23.56 | 0.831 | 0.144 |
| Ignatius | 22.46 | 0.715 | 0.211 | 23.45 | 0.727 | 0.197 | 22.52 | 0.715 | 0.211 | 23.49 | 0.727 | 0.196 | 24.11 | 0.743 | 0.187 | 23.93 | 0.739 | 0.190 | 23.92 | 0.734 | 0.194 |
| Playground | 20.51 | 0.613 | 0.384 | 20.94 | 0.619 | 0.375 | 21.02 | 0.622 | 0.370 | 21.39 | 0.629 | 0.358 | 21.52 | 0.632 | 0.380 | 21.41 | 0.629 | 0.383 | 21.82 | 0.634 | 0.374 |
| Train | 20.20 | 0.648 | 0.302 | 20.48 | 0.653 | 0.293 | 19.58 | 0.625 | 0.321 | 20.12 | 0.636 | 0.307 | 21.13 | 0.674 | 0.272 | 21.06 | 0.671 | 0.270 | 20.24 | 0.640 | 0.314 |
| Avg | 22.29 | 0.732 | 0.249 | 22.79 | 0.738 | 0.240 | 22.68 | 0.737 | 0.238 | 23.18 | 0.744 | 0.229 | 23.34 | 0.746 | 0.234 | 23.38 | 0.745 | 0.233 | 23.73 | 0.747 | 0.230 |

Table 14: **Quantitative results of novel view synthesis with camera pose optimization on LLFF dataset**.

| Scenes | COLMAP + 3DGS | | | COLMAP + 3DGS + pose opt | | | ACE0 | | | ACE0 + pose opt | | | Ours | | |
|---|---|---|---|---|---|---|---|---|---|---|---|---|---|---|---|
| | PSNR↑ | SSIM↑ | LPIPS↓ | PSNR↑ | SSIM↑ | LPIPS↓ | PSNR↑ | SSIM↑ | LPIPS↓ | PSNR↑ | SSIM↑ | LPIPS↓ | PSNR↑ | SSIM↑ | LPIPS↓ |
| Fern | 23.63 | 0.794 | 0.136 | 23.68 | 0.794 | 0.134 | 19.33 | 0.585 | 0.295 | 19.39 | 0.586 | 0.290 | 23.77 | 0.797 | 0.114 |
| Flower | 26.91 | 0.829 | 0.096 | 26.92 | 0.830 | 0.095 | 25.84 | 0.787 | 0.109 | 25.71 | 0.786 | 0.110 | 25.81 | 0.812 | 0.108 |
| Fortress | 29.93 | 0.880 | 0.078 | 29.92 | 0.880 | 0.077 | 29.09 | 0.848 | 0.085 | 29.21 | 0.849 | 0.085 | 29.31 | 0.868 | 0.084 |
| Horns | 26.02 | 0.862 | 0.121 | 25.72 | 0.854 | 0.130 | 24.99 | 0.821 | 0.137 | 24.93 | 0.819 | 0.142 | 26.67 | 0.882 | 0.093 |
| Leaves | 17.91 | 0.593 | 0.205 | 18.06 | 0.596 | 0.201 | 17.09 | 0.527 | 0.242 | 17.10 | 0.529 | 0.244 | 16.45 | 0.526 | 0.270 |
| Orchids | 18.98 | 0.612 | 0.159 | 18.99 | 0.613 | 0.160 | 16.15 | 0.419 | 0.314 | 16.12 | 0.417 | 0.316 | 19.55 | 0.640 | 0.147 |
| Room | 28.96 | 0.927 | 0.115 | 29.76 | 0.933 | 0.097 | 26.89 | 0.894 | 0.118 | 26.96 | 0.894 | 0.120 | 32.37 | 0.948 | 0.073 |
| Trex | 24.74 | 0.881 | 0.145 | 25.07 | 0.884 | 0.097 | 22.52 | 0.792 | 0.155 | 22.25 | 0.790 | 0.155 | 26.11 | 0.905 | 0.084 |
| Avg | 24.64 | 0.794 | 0.132 | 24.77 | 0.797 | 0.124 | 22.74 | 0.709 | 0.182 | 22.71 | 0.709 | 0.183 | 25.01 | 0.797 | 0.122 |

Table 15: **Quantitative results of novel view synthesis on MipNeRF360 dataset**.

| Scenes | COLMAP + 3DGS | | | COLMAP + 3DGS + pose opt | | | ACE0 + 3DGS | | | ACE0 + 3DGS + pose opt | | | Ours | | |
|---|---|---|---|---|---|---|---|---|---|---|---|---|---|---|---|
| | PSNR↑ | SSIM↑ | LPIPS↓ | PSNR↑ | SSIM↑ | LPIPS↓ | PSNR↑ | SSIM↑ | LPIPS↓ | PSNR↑ | SSIM↑ | LPIPS↓ | PSNR↑ | SSIM↑ | LPIPS↓ |
| Bicycle | 23.23 | 0.656 | 0.282 | 23.19 | 0.648 | 0.291 | 21.08 | 0.413 | 0.523 | 21.02 | 0.410 | 0.537 | 23.86 | 0.725 | 0.181 |
| Counter | 29.26 | 0.916 | 0.081 | 29.33 | 0.916 | 0.081 | 26.34 | 0.824 | 0.157 | 26.38 | 0.826 | 0.152 | 29.34 | 0.917 | 0.080 |
| Garden | 27.46 | 0.861 | 0.079 | 27.48 | 0.861 | 0.079 | 22.78 | 0.583 | 0.287 | 22.97 | 0.590 | 0.283 | 27.42 | 0.862 | 0.077 |
| Kitchen | 32.35 | 0.947 | 0.042 | 32.43 | 0.947 | 0.042 | 27.03 | 0.832 | 0.109 | 26.89 | 0.830 | 0.110 | 32.12 | 0.945 | 0.043 |
| Avg | 28.08 | 0.845 | 0.121 | 28.11 | 0.843 | 0.123 | 24.31 | 0.663 | 0.269 | 24.32 | 0.664 | 0.271 | 28.19 | 0.862 | 0.095 |

Table 16: **Quantitative results of novel view synthesis on Tanks-and-Temples dataset**.

| Scenes | COLMAP+3DGS | | | COLMAP+3DGS+pose opt | | | ACE0+3DGS | | | ACE0+3DGS+pose opt | | | Ours | | |
|---|---|---|---|---|---|---|---|---|---|---|---|---|---|---|---|
| | PSNR↑ | SSIM↑ | LPIPS↓ | PSNR↑ | SSIM↑ | LPIPS↓ | PSNR↑ | SSIM↑ | LPIPS↓ | PSNR↑ | SSIM↑ | LPIPS↓ | PSNR↑ | SSIM↑ | LPIPS↓ |
| Family | 23.04 | 0.773 | 0.147 | 23.08 | 0.773 | 0.145 | 20.43 | 0.651 | 0.259 | 19.32 | 0.591 | 0.352 | 23.11 | 0.775 | 0.144 |
| Francis | 26.42 | 0.841 | 0.188 | 26.88 | 0.846 | 0.178 | 24.08 | 0.796 | 0.274 | 24.18 | 0.793 | 0.275 | 26.70 | 0.845 | 0.182 |
| Ignatius | 22.07 | 0.671 | 0.228 | 21.99 | 0.669 | 0.229 | 20.93 | 0.615 | 0.273 | 20.92 | 0.614 | 0.271 | 21.95 | 0.669 | 0.230 |
| Train | 20.72 | 0.685 | 0.235 | 20.74 | 0.686 | 0.235 | 18.29 | 0.580 | 0.400 | 18.24 | 0.579 | 0.398 | 20.93 | 0.689 | 0.230 |
| Avg | 23.06 | 0.743 | 0.200 | 23.17 | 0.743 | 0.197 | 20.93 | 0.661 | 0.302 | 20.67 | 0.644 | 0.324 | 23.17 | 0.745 | 0.197 |

Table 17: **Quantitative results of novel view synthesis on 7scenes dataset**.

| Scenes | COLMAP+3DGS | | | COLMAP+3DGS+pose opt | | | ACE0+3DGS | | | ACE0+3DGS+pose opt | | | Ours | | |
|---|---|---|---|---|---|---|---|---|---|---|---|---|---|---|---|
| | PSNR↑ | SSIM↑ | LPIPS↓ | PSNR↑ | SSIM↑ | LPIPS↓ | PSNR↑ | SSIM↑ | LPIPS↓ | PSNR↑ | SSIM↑ | LPIPS↓ | PSNR↑ | SSIM↑ | LPIPS↓ |
| chess | 26.13 | 0.872 | 0.151 | 26.13 | 0.873 | 0.151 | 25.48 | 0.855 | 0.165 | 25.52 | 0.858 | 0.159 | 26.02 | 0.872 | 0.154 |
| fire | 26.02 | 0.805 | 0.193 | 26.17 | 0.807 | 0.187 | 25.61 | 0.774 | 0.201 | 25.38 | 0.763 | 0.210 | 25.67 | 0.805 | 0.192 |
| heads | 23.03 | 0.843 | 0.225 | 23.06 | 0.840 | 0.230 | 24.40 | 0.820 | 0.298 | 24.22 | 0.861 | 0.195 | 23.68 | 0.854 | 0.211 |
| pumpkin | 26.62 | 0.864 | 0.219 | 26.63 | 0.864 | 0.220 | 26.86 | 0.855 | 0.212 | 26.93 | 0.860 | 0.207 | 26.54 | 0.863 | 0.218 |
| Avg | 25.45 | 0.846 | 0.197 | 25.50 | 0.846 | 0.197 | 25.58 | 0.826 | 0.219 | 25.51 | 0.836 | 0.193 | 25.48 | 0.849 | 0.194 |

Table 18: **Quantitative results of camera pose accuracy on the sequential Tanks-and-Temples dataset**.

| Scenes | BARF ΔR | BARF Δt | NeRFmm ΔR | NeRFmm Δt | Nope-NeRF ΔR | Nope-NeRF Δt | CF-3DGS ΔR | CF-3DGS Δt | ACE0 ΔR | ACE0 Δt | Ours ΔR | Ours Δt |
|---|---|---|---|---|---|---|---|---|---|---|---|---|
| Ballroom | 0.228 | 0.531 | 0.177 | 0.449 | 0.018 | 0.041 | 0.024 | 0.037 | 8.047 | 0.018 | 1.303 | 0.002 |
| Barn | 0.265 | 0.314 | 0.494 | 1.629 | 0.032 | 0.046 | 0.034 | 0.034 | 10.488 | 0.035 | 0.161 | 0.001 |
| Church | 0.038 | 0.114 | 0.127 | 0.626 | 0.008 | 0.034 | 0.018 | 0.008 | 4.316 | 0.023 | 0.136 | 0.004 |
| Family | 0.591 | 1.371 | 0.537 | 2.743 | 0.015 | 0.047 | 0.024 | 0.022 | 1.589 | 0.029 | 0.111 | 0.002 |
| Francis | 0.558 | 1.321 | 0.618 | 1.647 | 0.009 | 0.057 | 0.154 | 0.029 | 18.370 | 0.026 | 0.429 | 0.002 |
| Horse | 0.394 | 1.333 | 0.434 | 1.349 | 0.017 | 0.179 | 0.057 | 0.112 | 2.439 | 0.023 | 0.216 | 0.007 |
| Ignatius | 0.324 | 0.736 | 0.379 | 1.302 | 0.005 | 0.026 | 0.057 | 0.122 | 2.434 | 0.025 | 0.133 | 0.002 |
| Museum | 1.128 | 3.442 | 1.051 | 4.134 | 0.202 | 0.207 | 0.215 | 0.052 | 28.510 | 0.648 | 0.227 | 0.010 |
| Avg | 0.441 | 1.046 | 0.477 | 1.735 | 0.038 | 0.080 | 0.069 | 0.041 | 9.524 | 0.103 | 0.313 | 0.002 |

Table 19: **Quantitative results of novel view synthesis on the sequential Tanks-and-Temples dataset**.

| Scenes | BARF PSNR↑ | BARF SSIM↑ | BARF LPIPS↓ | NeRFmm PSNR↑ | NeRFmm SSIM↑ | NeRFmm LPIPS↓ | Nope-NeRF PSNR↑ | Nope-NeRF SSIM↑ | Nope-NeRF LPIPS↓ | CF-3DGS PSNR↑ | CF-3DGS SSIM↑ | CF-3DGS LPIPS↓ | ACE0+3DGS PSNR↑ | ACE0+3DGS SSIM↑ | ACE0+3DGS LPIPS↓ | COLMAP+3DGS PSNR↑ | COLMAP+3DGS SSIM↑ | COLMAP+3DGS LPIPS↓ | Ours PSNR↑ | Ours SSIM↑ | Ours LPIPS↓ |
|---|---|---|---|---|---|---|---|---|---|---|---|---|---|---|---|---|---|---|---|---|---|
| Ballroom | 20.66 | 0.50 | 0.60 | 20.03 | 0.48 | 0.57 | 25.33 | 0.72 | 0.38 | 32.47 | 0.96 | 0.07 | 22.28 | 0.80 | 0.17 | 34.33 | 0.96 | 0.02 | 35.18 | 0.97 | 0.01 |
| Barn | 25.28 | 0.64 | 0.48 | 23.21 | 0.61 | 0.53 | 26.35 | 0.69 | 0.44 | 31.23 | 0.90 | 0.10 | 25.88 | 0.87 | 0.15 | 32.66 | 0.95 | 0.04 | 31.81 | 0.95 | 0.04 |
| Church | 23.17 | 0.62 | 0.52 | 21.64 | 0.58 | 0.54 | 25.17 | 0.73 | 0.39 | 30.23 | 0.93 | 0.11 | 28.47 | 0.89 | 0.09 | 30.04 | 0.93 | 0.06 | 29.80 | 0.93 | 0.06 |
| Family | 23.04 | 0.61 | 0.56 | 23.04 | 0.58 | 0.56 | 26.01 | 0.74 | 0.41 | 31.27 | 0.94 | 0.07 | 23.31 | 0.85 | 0.15 | 28.74 | 0.93 | 0.07 | 27.99 | 0.92 | 0.08 |
| Francis | 25.85 | 0.69 | 0.57 | 25.40 | 0.69 | 0.52 | 29.48 | 0.80 | 0.38 | 32.72 | 0.91 | 0.14 | 27.14 | 0.83 | 0.24 | 32.01 | 0.92 | 0.11 | 31.76 | 0.92 | 0.11 |
| Horse | 24.09 | 0.72 | 0.41 | 23.12 | 0.70 | 0.43 | 27.64 | 0.84 | 0.26 | 33.94 | 0.96 | 0.05 | 22.33 | 0.80 | 0.21 | 22.66 | 0.82 | 0.18 | 26.32 | 0.90 | 0.11 |
| Ignatius | 21.78 | 0.47 | 0.60 | 21.16 | 0.45 | 0.60 | 23.96 | 0.61 | 0.47 | 28.43 | 0.90 | 0.09 | 24.54 | 0.81 | 0.16 | 30.17 | 0.92 | 0.06 | 30.86 | 0.93 | 0.05 |
| Museum | 23.58 | 0.61 | 0.55 | 22.37 | 0.61 | 0.53 | 26.77 | 0.76 | 0.35 | 29.91 | 0.91 | 0.11 | 20.62 | 0.65 | 0.31 | 34.51 | 0.96 | 0.02 | 33.97 | 0.94 | 0.02 |
| Avg | 23.42 | 0.61 | 0.54 | 22.50 | 0.59 | 0.54 | 26.34 | 0.74 | 0.39 | 31.28 | 0.93 | 0.09 | 24.32 | 0.81 | 0.19 | 30.64 | 0.92 | 0.07 | 30.96 | 0.93 | 0.06 |

feed-forward manner. Regardless of its efficiency in inference, the accuracy of the reconstruction results is still not well solved. We believe that many variants of DUSt3R will benefit from our work in improving their reconstruction accuracy.

Table 20: **Quantitative results of novel view synthesis with camera pose optimization on the sequential Tanks-and-Temples dataset**.

| Scenes | COLMAP + 3DGS PSNR↑ | COLMAP + 3DGS SSIM↑ | COLMAP + 3DGS LPIPS↓ | COLMAP + 3DGS + pose opt PSNR↑ | COLMAP + 3DGS + pose opt SSIM↑ | COLMAP + 3DGS + pose opt LPIPS↓ | ACE0 + 3DGS PSNR↑ | ACE0 + 3DGS SSIM↑ | ACE0 + 3DGS LPIPS↓ | ACE0 + 3DGS + pose opt PSNR↑ | ACE0 + 3DGS + pose opt SSIM↑ | ACE0 + 3DGS + pose opt LPIPS↓ | Ours PSNR↑ | Ours SSIM↑ | Ours LPIPS↓ |
|---|---|---|---|---|---|---|---|---|---|---|---|---|---|---|---|
| Ballroom | 34.33 | 0.96 | 0.02 | 34.42 | 0.97 | 0.02 | 22.28 | 0.80 | 0.17 | 25.24 | 0.86 | 0.11 | 35.18 | 0.97 | 0.01 |
| Barn | 32.66 | 0.95 | 0.04 | 32.20 | 0.95 | 0.04 | 25.88 | 0.87 | 0.15 | 25.82 | 0.87 | 0.14 | 31.81 | 0.95 | 0.04 |
| Church | 30.04 | 0.93 | 0.06 | 29.95 | 0.93 | 0.06 | 28.47 | 0.89 | 0.09 | 28.32 | 0.90 | 0.09 | 29.80 | 0.93 | 0.06 |
| Family | 28.74 | 0.93 | 0.07 | 28.94 | 0.93 | 0.07 | 23.31 | 0.85 | 0.15 | 23.40 | 0.85 | 0.15 | 27.99 | 0.92 | 0.08 |
| Francis | 32.01 | 0.92 | 0.11 | 32.24 | 0.92 | 0.11 | 27.14 | 0.83 | 0.24 | 26.72 | 0.83 | 0.24 | 31.76 | 0.92 | 0.11 |
| Horse | 22.66 | 0.82 | 0.18 | 22.11 | 0.81 | 0.19 | 22.33 | 0.80 | 0.21 | 22.53 | 0.81 | 0.20 | 26.32 | 0.90 | 0.11 |
| Ignatius | 30.17 | 0.92 | 0.06 | 30.34 | 0.93 | 0.06 | 24.54 | 0.81 | 0.16 | 23.29 | 0.80 | 0.18 | 30.86 | 0.93 | 0.05 |
| Museum | 34.51 | 0.96 | 0.02 | 34.51 | 0.96 | 0.02 | 20.62 | 0.65 | 0.31 | 20.63 | 0.66 | 0.31 | 33.97 | 0.94 | 0.02 |
| Avg | 30.64 | 0.92 | 0.07 | 30.59 | 0.93 | 0.07 | 24.32 | 0.81 | 0.19 | 24.49 | 0.82 | 0.18 | 30.96 | 0.93 | 0.06 |

