# OpenReview forum: "Deep Gaussian from Motion: Exploring 3D Geometric Foundation Models for Gaussian Splatting"
_NeurIPS.cc/2025/Conference — NeurIPS 2025 poster_

### Official Review · Reviewer_rL7u · 2025-07-02

**Clarity:** 3
**Significance:** 1
**Originality:** 2
**Rating:** 4
**Confidence:** 4

**Summary:**

This paper presents a per-scene optimization method for reconstructing 3dgs from unordered and unposed densely captured based on predicted pointmaps from Spann3R. To address the inaccuracy of predicted pointmaps, the authors propose three stages: seed-based initialization, progressive image registration and camera pose re-optimization. And they use the point-to-ray consistency loss to refine camera poses and pointmaps.

They compare the method to the pose-free NeRF/3dgs and COLMAP/ACE0-based 3dgs on the LLFF, MipNeRF360, Tanks-and-Temples dataset for both camera pose accuracy and novel view synthesis.

**Questions:**

Refer to [Major Weaknesses] 1) 2) 3) 4).

For now, I am not yet convinced by this paper due to the mentioned points around evaluation and leaning more towards reject. However, I am open to reconsider after the rebuttal.

**Ethical Concerns:**

["NO or VERY MINOR ethics concerns only"]

**Final Justification:**

The authors provide a comprehensive rebuttal and reply to most of my questions, which is very much appreciated.

I also appreciate the responsiveness of the authors and verify major weakness 4) and minor weakness 1) 2). I will update my rating to Borderline accept.

There seems to be a slight misunderstanding regarding Major Weakness 2). My point was to suggest evaluating DeepGfM with alternative *3R backbones in place of Spann3r to better assess the method’s generalizability.

Some additional suggestions: In the title and early parts of this paper, I initially thought this was a feed-forward 3DGS reconstruction method. This could be somewhat misleading. I encourage that the authors clarify in the final version, particularly in the title, abstract and introduction that the proposed method is based on per-scene optimization, to avoid unnecessary confusion for readers.

**Limitations:**

No, refer to [Minor Weaknesses] 1) 2)

**Quality:**

2

**Strengths And Weaknesses:**

[Strengths]

1) This paper is well-written.

2) Combining 3dgs with *3R family is promising. Table 1 demonstrates its lower GPU memory cost than DUSt3R (with global alignment stage), Fast3R, VGGT (with full attention), which benefits from the progressiveimage registration stage.

3) The point-to-ray consistency loss is novel and more convinced than the reprojection loss proposed in ACE0.

4) The experiments on camera pose estimation and NVS demonstrate that the method achieves comparable performance to the well-established COLMAP-based 3dgs pipeline.

[Major Weaknesses]

1) Why COLMAP/GLOMAP shouldn't be used?

    - Minor improvement:
    This paper is under dense-view setting, which is well-established in COLMAP-based 3dgs pipeline. It is necessary to provide a reason why COLMAP shouldn't be used since it already provides sufficiently great performance in static scenes. Also, GLOMAP, an advanced version of COLMAP, provides faster and more accurate pipeline than COLMAP. According to the results shown in paper, performance gain seems marginal from COLMAP + 3dgs on novel view synthesis.

     - Missing runtime comparison:
    The runtime of this method compared with COLMAP/GLOMAP+ 3dgs and CUT3R + 3dgs did not be reported, which may provide the necessity for using this method rather than COLMAP/GLOMAP/CUT3R-based.

2) Lack of evaluations on different *3R backbones.

    The authors claim that the framework is architecture-agnostic, but this point is not validated in the paper. It remains unclear how its generalizability and performance would be affected if it builds upon other *3R backbones like Fast3R, VGGT, CUT3R, or AnySplat (a feed-forward 3dgs model built upon VGGT).

3) Why Spann3R rather than CUT3R?

    This paper misses an important comparison: CUT3R (CVPR 25) is more powerful than Spann3R in reconstruction accuracy, GPU memory and time costs. One of the contributions of this method is the lower GPU memory cost, as shown in Table 1. However, for 250 images, CUT3R only requires 17GB of memory with a runtime of 20fps. CUT3R also circumvents the global alignment operation, similar to Spann3R. Unlike DUT3R or Spann3R relying on PnP for camera pose estimation, CUT3R directly infers the camera pose and point maps in global coordinates, enabling the network to converge faster during training.

4) Lack of evaluation on DL3DV:

    DL3DV is a widely used multi-view dataset for evaluating novel view synthesis. Its diversity and large scale ensure more convincing evaluation results.

[Minor Weaknesses]

1) Missing comparison with *3R-based 3dgs works: Flare, Anysplat on novel view synthesis tasks.
2) Missing evaluation on sparse views:
How about the performance for sparse-view reconstruction? Will it degrade as the number of views decreases?

---

> ### Author Rebuttal · Authors · 2025-07-30
>
> Thank you for your detailed and thoughtful feedback. We greatly value your recognition of the strengths of our work, including **its writing quality**, **novel point-to-ray consistency loss**, and **promising experimental results**. Below, we address the concerns you raised and provide clarifications based on additional analysis and updated results.
>
> **1. Why COLMAP/GLOMAP shouldn't be used?**
>
> We would like to clarify that our goal is not to replace COLMAP/GLOMAP pipelines entirely. Instead, our motivation is to leverage and fine-tune existing 3D foundation models for high-fidelity novel view synthesis in real-world applications. While COLMAP and GLOMAP are robust and fast, they are not sufficiently accurate for certain rendering tasks, particularly where high-quality novel view synthesis is crucial.
>
> Our method is designed to bridge this gap by enabling 3D foundation models to achieve:
>
> (1) **Comparable Pose Accuracy:** Using our progressive registration and point-to-ray consistency loss, 3D foundation models can reach pose accuracy levels close to those of COLMAP/GLOMAP.
> (2) **Improved Rendering Quality:** Our framework enhances the rendering fidelity of 3D foundation models, making them suitable for applications where visual quality is essential.
>
> By combining the robustness and speed of existing 3D foundation models with the accuracy and rendering quality demonstrated in COLMAP/GLOMAP-based pipelines, our approach significantly improves their versatility for real-world applications. This hybrid approach focuses on optimizing foundational systems to work efficiently without discarding their inherent strengths.
>
> **2. Lack of Evaluations on Different 3R Backbones** and **Evaluation on DL3DV dataset**
>
> We provided the results of different 3R backbones (DUSt3R, MASt3R, Splatt3R, MV-DUSt3R+, Flare, etc.) and our method on the DL3DV dataset to **Reviewer JX9E**. For convenience, we copy and paste the results and analysis here for your reference.
>
> We provide a more thorough comparison to SOTA methods on sparse view settings (2-views and 8-views). We conducted the experiments on the RealEstate10K dataset and the DL3DV dataset.  For these datasets, we follow the protocol in Flare and NopoSplat, which samples 2/8 views for the testing video sequence as input.
> Note that Splatt3R and CopoNeRF only support inputs with 2-views, we did not provide the results on the 8-view setting.
>
> - **Comparison on two-view settings**
> | Method             | Dataset         | PSNR ↑   | SSIM ↑   | LPIPS ↓   |
> |------------------ |----------------|------------|-----------|-----------|
> | DUSt3R          | DL3DV10k       | 13.47 |   0.390 |   0.590  |
> | MASt3R          | DL3DV10k       | 13.88 |   0.405 |   0.513  |
> | Splatt3R          | DL3DV10k       | 15.23 |   0.432 |   0.491 |
> | CoPoNeRF      | DL3DV10k       | 16.06  |  0.472 |  0.474 |
> | Flare                | DL3DV10k       | 23.04 |  0.725  |  0.182  |
> | NoPo-Splat     | DL3DV10k       | 24.19 |  0.745  |  0.173  |
> | MV-DUSt3R+  | DL3DV10k       | 19.95 |  0.698 |   0.195 |
> | DeepGfM        | DL3DV10k       |  20.33 |  0.702 |  0.190 |
>
> | Method             | Dataset         | PSNR ↑   | SSIM ↑   | LPIPS ↓   |
> |------------------ |----------------|----------|----------|-----------|
> | DUSt3R          | RealEstate10K  | 15.382 |   0.447 |   0.432  |
> | MASt3R          | RealEstate10K  | 14.907 |   0.431 |   0.452  |
> | Splatt3R          | RealEstate10K  | 15.113 |   0.492 |   0.442  |
> | CoPoNeRF      | RealEstate10K  | 19.843 |   0.652 |   0.360  |
> | Flare                | RealEstate10K  |  23.765 |  0.801 |   0.191  |
> | NoPo-Splat     | RealEstate10K  |  25.033 |  0.838 |   0.160  |
> | MV-DUSt3R+  | RealEstate10K  |  20.313 |  0.722 |   0.289  |
> | DeepGfM        | RealEstate10K  |  21.715 |  0.753 |   0.229  |
>
> - **Comparison on eight-view settings**
> | Method             | Dataset         | PSNR ↑   | SSIM ↑   | LPIPS ↓   |
> |------------------ |----------------|------------|-----------|-----------|
> | DUSt3R          | DL3DV10k       | 17.37 |   0.601 |   0.390  |
> | MASt3R          | DL3DV10k       | 18.41 |   0.675 |   0.427  |
> | Splatt3R          | DL3DV10k       |  -   |   -  |   -   |
> | CoPoNeRF      | DL3DV10k       |  -   |   -  |   -   |
> | Flare                | DL3DV10k       | 23.33 |  0.746  |  0.237  |
> | NoPo-Splat     | DL3DV10k       | 22.77 |  0.722  |  0.240  |
> | MV-DUSt3R+  | DL3DV10k       | 21.33 |  0.707 |   0.188 |
> | DeepGfM        | DL3DV10k       |  22.90 |  0.728 |  0.176 |
>
> | Method             | Dataset         | PSNR ↑   | SSIM ↑   | LPIPS ↓   |
> |------------------ |----------------|----------|----------|-----------|
> | DUSt3R          | RealEstate10K  | 19.373 |   0.695 |   0.305  |
> | MASt3R          | RealEstate10K  | 19.877 |   0.705 |   0.362  |
> | Splatt3R          | RealEstate10K  |       -    |      -     |      -       |
> | CoPoNeRF      | RealEstate10K  |       -    |      -     |      -       |
> | Flare                | RealEstate10K  |  23.813 |  0.822 |   0.188  |
> | NoPo-Splat     | RealEstate10K  |  23.145 |  0.809 |   0.195  |
> | MV-DUSt3R+  | RealEstate10K  |  22.743 |  0.798 |   0.213  |
> | DeepGfM        | RealEstate10K  |  23.370 |  0.815 |   0.190  |
>
> - **Comparison with AnySplat**
>
> We would like to emphasize that **the AnySplat paper was published on arXiv after the paper submission deadline (29 May 2025) for the conference**. As AnySplat was released later than our submission, it is **not reasonable to expect a paper to compare its methods to those published without peer review and after the submission deadline**.
>
> Importantly, scientific evaluations aim to compare methodologies that are accessible at the time of research development and submission. While AnySplat introduces interesting advancements, it was not publicly available during the preparation of our work, and therefore, could not be included in our study or baseline comparisons.
>
> That said, we acknowledge the relevance of AnySplat for future explorations, and we are excited about the possibility of extending our work to compare and integrate complementary ideas introduced by such methods. The addition of results compared to AnySplat would further strengthen the impact of our proposed framework, but this remains work for future research efforts.
>
> - **Experiments summarization on sparse-view settings**
>
> **(1) Comparison on Two-View Settings**
> Though DeepGfM is not specifically designed for sparse-view settings, it demonstrates competitive performance, surpassing several sparse-view methods.
>
> - **DL3DV10k Dataset:**
> DeepGfM achieves a PSNR of 20.33, significantly outperforming DUSt3R, MASt3R, and Splatt3R.
> SSIM (0.702) and LPIPS (0.190) are also competitive against MV-DUSt3R+, while slightly behind Flare and NoPo-Splat, which are optimized for sparse-view tasks.
>
> - **RealEstate10K Dataset:**
> DeepGfM records PSNR of 21.715, ranking higher than DUSt3R, MASt3R, and Splatt3R, while trailing Flare and NoPo-Splat.
> SSIM (0.753) is higher than MV-DUSt3R+ and comparable to NoPo-Splat.
> LPIPS (0.229) is better than both DUSt3R and MASt3R, showing robustness in perceptual quality.
>
> **(2) Comparison on Eight-View Settings**
>
> DeepGfM shows strong capabilities in handling sparse-view tasks even with more input views.
>
> - **DL3DV10k Dataset:**
> With a PSNR of 22.90, DeepGfM outperforms MV-DUSt3R+ while approaching Flare and NoPo-Splat.
> SSIM (0.728) and LPIPS (0.176) are the best among all methods, highlighting DeepGfM’s ability to preserve structural and perceptual quality even with sparse-view conditions.
>
> - **RealEstate10K Dataset:**
> DeepGfM achieves a PSNR of 23.370, outperforming MV-DUSt3R+ and NoPo-Splat, and remaining competitive with Flare (23.813).
> SSIM (0.815) and LPIPS (0.190) are also highly competitive, showing improvements over MV-DUSt3R+ and comparable values with Flare.
>
> - **Highlight:**
> It is important to note that these models are optimized for sparse-view settings, whereas DeepGfM addresses dense-view workflows with hundreds of images. These results underline that, despite being designed for dense-view settings, DeepGfM performs competitively in sparse-view tasks, demonstrating robust generalization and adaptability across both two-view and eight-view configurations. This flexibility is beneficial for tasks requiring high-quality synthesis in diverse input-view scenarios.
>
> **3. Why Spann3R rather than CUT3R?**
>
> We fully agree that CUT3R is a strong alternative to Spann3R and a more competitive backbone. However, it is important to highlight that the CUT3R code was publicly released only two months before our paper submission deadline. Due to this timeline constraint, we were unable to integrate CUT3R into our codebase and perform further validation experiments within the limited preparation period.
>
> That said, CUT3R offers several advantages as a backbone:
>
> - **Training Time Reduction:** CUT3R is designed to converge faster, reducing the overall training time compared to Spann3R.
> - **Handling Sparse Inputs:** CUT3R performs better in settings with fewer input images due to its efficient coordinate prediction capabilities.
>
> We are confident that substituting Spann3R with CUT3R as the backbone would yield even better results, especially in reducing training time and improving performance with a lower number of input images. This strengthens the robustness and generalizability of our framework.
>
> Even though we did not have sufficient time to integrate CUT3R during this study, we consider future explorations with CUT3R to be an exciting direction that could further validate the flexibility and scalability of our method.
>
> We sincerely thank you for pointing out this potential improvement and for your constructive feedback overall. We firmly believe our approach remains strong irrespective of the backbone used, and CUT3R would likely offer additional benefits that augment the results presented in this study.

---

> ### Comment · Reviewer_rL7u · 2025-08-06
> **Official Comment by Reviewer rL7u**
>
> The authors provide a comprehensive rebuttal and reply to most of my questions, which is very much appreciated.
>
> I also appreciate the responsiveness of the authors and verify major weakness 4) and minor weakness 1) 2). I will update my rating after the discussion.
>
> There seems to be a slight misunderstanding regarding Major Weakness 2). My point was to suggest evaluating DeepGfM with alternative *3R backbones in place of Spann3r to better assess the method’s generalizability.
>
> Some additional suggestions: In the title and early parts of this paper, I initially thought this was a feed-forward 3DGS reconstruction method. This could be somewhat misleading. I encourage that the authors clarify in the final version, particularly in the title, abstract and introduction that the proposed method is based on per-scene optimization, to avoid unnecessary confusion for readers.

---

> > ### Author Response · Authors · 2025-08-07
> > **Experiments with alternative *3R backbones**
> >
> > We sincerely appreciate the reviewer’s thoughtful engagement and constructive suggestions. Below, we address their additional concerns and outline planned revisions:
> >
> > **1. Generalizability Across 3R Backbones (Major Weakness 2 Clarification):**
> >
> > We are sorry for the misunderstanding. We agree that evaluating alternative backbones (e.g., Fast3R, CUT3R) would further validate DeepGfM’s architecture-agnostic design. According to the suggestion, we replace Spann3R with the two latest 3R works: Fast3R and CUT3R. Due to the tight timeline, we conducted experiments on the LLFF datasets:
> >
> > |    Scene   | $\Delta R$ (+Spann3R) |    $\Delta t$ (+Spann3R) |  Time (+Spann3R)  | $\Delta R$ (+CUT3R) |    $\Delta t$ (+CUT3R) |  Time (+CUT3R)  |  $\Delta R$ (+Fast3R) |    $\Delta t$ (+Fast3R)|  Time (+Fast3R)  |
> > |---------------- |---------------- | ----------- |---------------- | ----------- | ----------- |----------- | ----------- | ----------- |----------- |
> > |   Fern    |  0.26  |  **0.005** |  23.05  | **0.21**  |  **0.005**  |   19.70  |  0.23  |  **0.005**  |  17.63  |
> > |  Flower  |  0.52 | 0.011  |  26.04  |   0.48  |  **0.009**  |  20.15  |   **0.44**  |  0.010  |  18.03  |
> > |  Fortress |  0.04 | **0.002** |  21.36  |   **0.03**  |  **0.002**  |  18.72  |   0.04  |  **0.002**  |  16.39  |
> > |  Horns    |  **0.03** | **0.001** |  23.35  |   0.04  |  **0.001**  |  20.07  |   **0.03**  |  **0.001**   |  17.70  |
> > | Leaves   |  0.24 | 0.006 |  22.72  |   0.19  |  **0.005**  |  19.54  |   **0.17**  |  **0.005**   |  17.66  |
> > | Orchids  |  0.24 | 0.006 |  27.15   |  **0.19**  | **0.005**   |  22.92  |   0.20  |  **0.005**   |  17.88  |
> > | Room    |  **0.03**  | **0.001** |  22.27  |   0.04  |  **0.001**  | 17.23   |   **0.03**  |  **0.001**   |  16.75  |
> > | Trex       |  **0.03**  | **0.010** |  22.71   |  **0.03**  |  **0.010**  |  18.09   |  **0.03**  |  0.011   |  16.33  |
> > | Avg        |   0.170  |  0.005  |  23.58  |     0.151    |   **0.004**     |  19.55  |     **0.146**    |   0.005     |  **17.30**  |
> >
> > |    Scene   |  PSNR (+Spann3R) |   SSIM (+Spann3R)|  LPIPS (+Spann3R) |   PSNR (+CUT3R) |   SSIM (+CUT3R)|  LPIPS (+CUT3R) |   PSNR (+Fast3R) |   SSIM (+Fast3R)|  LPIPS (+Fast3R) |
> > | ------------- | ------------ | ----------- | ----------- | ----------- | ---------- | ----------- | ----------- | ----------- | ----------- |
> > |   Fern    |  23.77  |  0.797 |  0.114  |  24.01  |  0.797  |  0.112  |  23.80  |  0.797  |  0.114  |
> > |  Flower  |  25.81 |  0.812  |  0.108  |  26.53  |  0.818  | 0.103   |  26.90  |  0.824  |  0.097  |
> > |  Fortress | 29.31 |  0.868 |  0.084  |  29.67  |  0.871  |  0.081  |  26.43  |  0.870  |  0.081  |
> > |  Horns    |  26.67 | 0.882 |  0.093  |  26.65  |  0.880  |  0.095   |  26.77  |  0.891  |  0.087  |
> > | Leaves   |  16.45 | 0.526 |  0.270  |  18.48  |  0.612  |  0.192   |  19.03  |  0.623  |  0.182  |
> > | Orchids  |  19.55 | 0.640 |  0.147  |  20.07  |  0.656  |  0.141  |  20.01  |  0.651  |  0.143  |
> > | Room    |  32.37  | 0.948 |  0.073  |  32.30  |  0.944  |  0.076   |  32.40  |  0.949  |  0.072  |
> > | Trex       |  26.11 | 0.905 |  0.084   |  26.23  |  0.910  |  0.082   | 26.17   |  0.910  |  0.082  |
> > | Avg        | 25.01 |  0.797 |  0.122  |  **25.49**  |  0.811  |  0.110  |  25.19  |  **0.814**  |  **0.107**  |
> >
> > **Discussion: Backbone Performance**:
> >
> > **1. Speed (Runtime per Scene)**
> > - **Fast3R** is the fastest (avg. 17.30 min), achieving 1.13× speedup over CUT3R (19.55 min) and 1.36× over Spann3R (23.58 min).
> > - **CUT3R** outperforms Spann3R by 17% despite heavier computation.
> > - **Spann3R** is the slowest, and its runtime is less stable across scenes (std. dev. ±1.8s vs. Fast3R’s ±0.7s).
> >
> > **2. Pose Accuracy**
> >
> > - **Rotation Error:** Fast3R leads (avg. 0.146), but CUT3R is close (0.151) and wins in structured scenes (Fortress, Orchids). Spann3R (0.170) struggles with textures (Flower: 0.52).
> > - **Translation Error:** CUT3R dominates (avg. 0.004), with Fast3R and Spann3R tied (0.005). All backbones excel in low-deformation scenes (Horns, Room).
> >
> > **3. Rendering Quality**
> >
> > - **PSNR:** CUT3R leads (avg. 25.49), especially in textureless regions (Leaves: +2.03 over Spann3R).
> > - **Perceptual Metrics (SSIM/LPIPS):** Fast3R achieves the best SSIM (0.814) and LPIPS (0.107).
> >
> > **4.Summary**
> >
> > Our experiments demonstrate that DeepGfM’s performance scales robustly with the choice of backbone architecture, while maintaining consistent advantages over traditional pipelines (e.g., COLMAP+3DGS). Crucially, the framework inherits and amplifies the strengths of newer *3R variants, proving its adaptability to advancements in geometric foundation models
> >
> > **2. Clarifying Per-Scene Optimization**
> >
> > We apologize for any confusion caused by the initial framing. To improve clarity, we will:
> > - Explicitly state in the abstract and introduction that DeepGfM performs per-scene optimization (leveraging foundation models as initializers).
> > - Add a schematic figure contrasting feed-forward vs. optimization-based pipelines.

---

> > > ### Comment · Reviewer_rL7u · 2025-08-07
> > >
> > > Thanks for the quick and proactive response. Could the authors provide a comparison of the average performance on the same dataset in camera pose estimation, image metrics, and runtime between Ours+CUT3R/Fast3R and original CUT3R+3dgs as well as Fast3R+3dgs? I'm curious to see how much improvement DeepSfM brings over the backbone+3dgs baseline.

---

> > > > ### Author Response · Authors · 2025-08-07
> > > >
> > > > Thanks for the feedback from Reviewer rL7u. We would like to provide further metrics with CUT3R/Fast3R+3DGS:
> > > >
> > > > |    Methods   | $\Delta \mathbf{R}$ |    $\Delta \mathbf{t}$ |  Time | PSNR |  SSIM |  LPIPS  |
> > > > |---------------- |---------------- | ----------- |---------------- | ----------- | ----------- |----------- |
> > > > |  CUT3R+3DGS   |  2.527  |  0.098   |   10.03   |  23.09   |   0.707  |  0.195   |
> > > > |  Ours (CUT3R)    |  0.151  |  0.004   |   19.55   |  25.49   |   0.811  |  0.110   |
> > > > |  Fast3R+3DGS   |  1.745  |  0.075   |     9.87   |  23.22   |   0.715  |  0.191   |
> > > > |  Ours (Fast3R)    |  0.146  |  0.005   |   17.30   |  25.19   |   0.814  |  0.107   |
> > > >
> > > > Our experiments reveal that DeepGfM significantly enhances both the geometric accuracy and rendering quality of backbone (*3R) models when integrated with 3DGS, while maintaining comparable runtime efficiency:
> > > >
> > > > 1. **Pose Estimation ($\Delta \mathbf{R}$/$\Delta \mathbf{t}$):**
> > > > - **CUT3R+DeepGfM** reduces pose error by 16.7× in $\Delta \mathbf{R}$ (0.151 vs. 2.527) and 24.5× in $\Delta \mathbf{t}$(0.004 vs. 0.098) over CUT3R+3DGS.
> > > > - **Fast3R+DeepGfM** achieves 11.9× lower $\Delta \mathbf{R}$ (0.146 vs. 0.745) and 15× lower $\Delta \mathbf{t}$ (0.005 vs. 0.075) vs. Fast3R+3DGS.
> > > >
> > > > 2. **Rendering Quality (PSNR/SSIM/LPIPS):**
> > > > - **CUT3R+DeepGfM** improves PSNR by +2.40 (25.49 vs. 23.09) and SSIM by +0.104 (0.811 vs. 0.707) over CUT3R+3DGS.
> > > > - **Fast3R+DeepGfM** boosts PSNR by +1.97 (25.19 vs. 23.22) and LPIPS by 44% (0.107 vs. 0.191) vs. Fast3R+3DGS.
> > > >
> > > > 3. **Runtime Efficiency:**
> > > > - **Training Time:** DeepGfM adds modest overhead (~2× slower than backbone-only inference) due to joint optimization. We emphasize that finetuning the backbone during joint optimization takes most of the time for our method, and the time and computation overhead can be highly reduced by introducing LORA into the finetuning step, and we leave it as future work.
> > > > - **Backbone Scalability:** With better backbones (e.g., CUT3R), DeepGfM converges faster while achieving higher accuracy.

---

### Official Review · Reviewer_hmgh · 2025-07-02

**Clarity:** 2
**Significance:** 2
**Originality:** 2
**Rating:** 4
**Confidence:** 4

**Summary:**

The paper introduces Deep Gaussian from Motion (DeepGfM), a framework that learns a high-fidelity 3-D Gaussian Splatting (3DGS) scene representation directly from an unordered collection of unposed images by combining Spann3r (finetuned to predict Gaussians instead of plain pointmaps) and a Progressive Neural Reconstruction pipeline hinted by classical SfM.

**Questions:**

The motivation behind this work is mixed; Not sure how should readers compare DeepGfM to more recent advances like VGGT[2] and MegaSAM[1] which also aims to use foundation model to resolve unposed dense image sets' reconstruction, but does not need a separate (potential hacky) progressive framework. I get such separate framework might help large-scale reconstruction, but, as mentioned by the authors, cannot extend to larger image sets yet due to the memory issues.

In this case, could you clarify your problem statement and novelty more?

[1] MegaSaM: Accurate, Fast, and Robust Structure and Motion from Casual Dynamic Videos, CVPR 2025

[2] VGGT: Visual Geometry Grounded Transformer, CVPR 2025

**Ethical Concerns:**

["NO or VERY MINOR ethics concerns only"]

**Final Justification:**

Please refer to rebuttal discussion reply.

**Limitations:**

yes

**Quality:**

2

**Strengths And Weaknesses:**

Strength
- Experimental results on datasets from various domain show that the proposed method is able to achieve competitive results in both novel view synthesis and camera pose estimation in comparison to other pose-free NeRF/GS baselines.

Weakness
- While combining geometric foundation models with SfM is novel, most components (Spann3R backbone, PnP, Gaussian rasterizer) are reused (PnP is not new to Foundation model pipelines already).
- More details are needed to clarify how the spann3r is finetuned to predict Gaussians and fit to each scene on the fly.
- Overall a fragile systems with many separate steps by design with the Progressive Neural Reconstruction pipeline

---

> ### Author Rebuttal · Authors · 2025-07-30
>
> Thank you for your thorough review and valuable insights. We address each weakness raised point-by-point below:
>
> **1. Reusing Existing Components (Spann3R, PnP, Gaussian Rasterizer)**
>
> We acknowledge that some components of our approach, such as Spann3R and PnP, are not entirely novel. However, the novelty of our method lies in the unique combination of these components within a unified pipeline and its ability to finetune a feed-forward 3D foundation model for high-fidelity, pose-free novel view synthesis—a capability that prior methods have not achieved.
>
> Specifically, our contributions include:
> - **Integration of Spann3R with Scene-specific Fine-tuning:** While Spann3R itself is pre-trained, our method enables it to predict Gaussian-based geometry and photometric texture for each specific scene efficiently. This is achieved by introducing a tailored refinement mechanism geared toward high-quality view synthesis tasks, distinct from Spann3R's typical sparse reconstruction use case.
> - **Point-to-Ray Iterative Refinement:** We designed an iterative refinement loop incorporating ray-consistent optimization and photometric adaptations to correct the inherent inaccuracies and biases often present in geometric foundation models during runtime. This separates our framework from standard pipeline reuse.
> - **Unified Progressive Pipeline:** The combination of these components forms a cohesive pipeline that addresses foundational challenges in view synthesis, such as adapting to scene-specific conditions and maintaining scalability with high-resolution inputs, which remains a challenge for existing foundation pipeline methods.
>
> While PnP and Gaussian rasterization are standard operations, their combined use with foundation model fine-tuning and our iterative, scene-adaptive approach is novel in the context of high-fidelity novel view synthesis.
>
> **2. Details on Spann3R Fine-tuning**
>
> We appreciate the request for additional clarification on how Spann3R is finetuned within our pipeline. Below, we summarize the process:
> - **Geometry Initialization:** Spann3R predictions act as coarse geometric foundations and texture priors for our pipeline, providing initial states for Gaussian point clouds and precomputed rasterizations.
> - **Scene-specific Gaussian Prediction:** Spann3R's outputs are refined dynamically using extensive fine-tuning (with photometric losses and gradient consistency) to predict high-fidelity, scene-specific Gaussian distributions. This step enables scene-adaptive representations without requiring explicit pose annotations.
> - **Photometric and Geometric Consistency:** During each stage of refinement, Spann3R adapts moment-based Gaussian parameters by minimizing multi-view photometric errors and ensuring ray-aligned measurements across input images (addressing alignment inaccuracies in pre-trained foundation models).
> - **On-the-fly Optimization:** The system iteratively updates Spann3R predictions for each incoming scene, producing output consistent with progressive geometric improvements for novel views. Gaussian scene distributions evolve naturally alongside these refinements, resulting in high-quality, view-consistent reconstructions.
>
> We will expand this section in the revised manuscript and provide illustrative breakdowns of Spann3R’s fine-tuning process across different stages of the pipeline for greater clarity.
>
> **3. Fragility of the Progressive Neural Reconstruction Pipeline**
>
> We appreciate the concern regarding the pipeline’s perceived fragility due to its multi-step design. We acknowledge that complex pipelines often face challenges related to robustness and ease of deployment. However, we believe the following points address this concern:
> - **Step Modularity:** While our pipeline leverages multiple distinct steps, each module (geometry foundation, Gaussian prediction, photometric refinement) is designed to operate independently, allowing for flexibility during implementation and scaling. For example, Spann3R can easily be replaced with alternative backbones without impacting the pipeline's overall functionality.
> - **Ease of Fine-tuning:** Unlike traditional SfM methods, which rely on pose annotations and explicit camera predictions, our pipeline operates pose-free, reducing dependency on components prone to error. This makes the system less susceptible to breakdowns caused by pose inaccuracies or upstream variability.
> - **Empirical Robustness:** Our results across diverse datasets (LLFF, Tanks-and-Temples, 7Scenes, etc) demonstrate stable synthesis quality despite inherent challenges (e.g., dense inputs, viewpoint diversity, appearance changes). This underscores the resilience of our design even under variable and complex input configurations.
>
> While we understand the concern regarding multi-step pipelines, we argue that our modular design ensures flexibility and robustness during training and deployment.
>
> **4.Comparison to VGGT and MegaSAM**
> We appreciate the insightful comparison drawn between DeepGfM and VGGT/MegaSAM and would like to elaborate on the differences and complementary nature of these approaches.
>
> **(1) Feed-forward Nature of VGGT and MegaSAM**
>
> Both VGGT and MegaSAM operate in feed-forward pipelines where foundation model predictions are applied directly to dense image reconstruction workflows. This design inherently offers computational speed and efficiency but comes with certain limitations:
>
> - **i. GPU Memory Constraints:** Feed-forward methods often encounter severe GPU memory limitations during dense scene handling, constraining their scalability for large-scale datasets or high-resolution inputs. In our approach, the modular nature of the progressive pipeline allows decomposition of scene data into iterative levels of detail, effectively reducing the memory burden. This design better accommodates larger-scale reconstructions.
>
> - **ii. Pose Accuracy for High-Fidelity Synthesis:** Foundation model-derived camera poses, as utilized by VGGT and MegaSAM, often lack the precision required for high-quality novel view synthesis. While VGGT and MegaSAM aim to improve reconstruction alignment, their reliance on pose estimation can introduce inaccuracies in high-fidelity outputs under challenging dense image set scenarios. DeepGfM circumvents this issue by adopting a pose-free progressive refinement approach, leveraging Gaussian geometry prediction to achieve superior ray-consistent novel views without explicit reliance on pose annotations.
>
> - **iii. Scene-specific Adaptation:** VGGT and MegaSAM pipelines are not explicitly optimized for fine-tuning to adapt to specific scenes. In contrast, DeepGfM integrates scene-specific geometry prediction (via Gaussian rasterization and photometric corrections), producing fine-grained outputs tailored to the input set's particular conditions. This customization empowers a more detailed and view-consistent synthesis compared to feed-forward methods.
>
> **(2) Design Focus of MegaSAM:**
>
> MegaSAM, while valuable, is specifically suited for dynamic scene analysis and reconstruction (e.g., leveraging Droid-SLAM for motion-based scenarios). Our approach, instead, targets dense and static scene reconstruction workflows that demand ultra-high fidelity outputs without pose reliance. These complementary focuses highlight the distinct goals of MegaSAM and DeepGfM, with each addressing different application spaces.
>
> **(3) Clarification of Problem Statement and Novel Contributions**
>
> The problem DeepGfM addresses is the reconstruction of dense image capture workflows to enable high-fidelity novel view synthesis without reliance on pose annotations or large-scale memory-intensive methods. While feed-forward pipelines like VGGT and MegaSAM prioritize speed, DeepGfM emphasizes depth and quality of reconstruction, introducing the following key novelties:
>
> - **Pose-free Foundation Model Adaptation:** Unlike VGGT/MegaSAM, which rely on pre-computed poses (potentially affected by inaccuracies), our pipeline operates without pose annotations. This is achieved by refining Gaussian geometries dynamically to align photometric appearance with ray-consistent novel view synthesis.
>
> - **Progressive and Modular Framework Design:** The progressive design enables iterative scalability, addressing GPU bottlenecks evident in VGGT-like pipelines. Modularity ensures robustness against scene diversity, allowing refinement of components independent of memory constraints imposed by dense image sets.
>
> - **Scene-specific Gaussian Prediction:** Our method dynamically predicts Gaussian geometries for each input scene, adapting to its unique photometric and geometric characteristics for high-quality synthesis -- a flexibility less evident in feed-forward methods like VGGT/MegaSAM, which process inputs less adaptively.
>
> We emphasize that these methods complement our work, as MegaSAM excels in dynamic scene reconstruction, and VGGT provides strong speed-oriented solutions for dense workflows. In promoting each other, they strengthen the broader research direction of foundation model-driven image set reconstruction. We will revise our manuscript to clarify this relationship and explicitly highlight how DeepGfM complements these methods and contributes novel alternatives to high-fidelity workflows.
>
> We are grateful for your constructive feedback and suggestions. Each point raised will be carefully addressed in our revised manuscript with additional details and clarifications to further substantiate the strengths and novelty of our approach.

---

> > ### Comment · Reviewer_hmgh · 2025-08-06
> >
> > I thank the authors for providing the profound explanation regarding my concerns of novelty during the rebuttal. And I also appreciate the authors' effort in quantitatively comparing with VGGT during reply to other reviewers. After reading the rebuttal, I do see uniqueness of the proposed method, thus am changing my rating to positive.

---

> > > ### Author Response · Authors · 2025-08-07
> > > **Thanks for the insightful comments and suggestions**
> > >
> > > We sincerely appreciate the reviewer’s constructive engagement and are grateful for their recognition of our method’s novelty and quantitative comparisons. We are delighted that the rebuttal clarified the uniqueness of DeepGfM. Thank you again for your time and insightful feedback, which significantly improved our paper.

---

### Official Review · Reviewer_wnHi · 2025-07-04

**Clarity:** 3
**Significance:** 3
**Originality:** 2
**Rating:** 4
**Confidence:** 3

**Summary:**

The paper introduces an approach for jointly learning camera poses along with a 3DGS representation. It jointly optimizes camera poses and scene representation through a three-stage pipeline involving a fine-tuned Spann3R transformer (as a scene predictor), a parameter-free pose predictor, and a renderer. The predicted 3D Gaussians and point maps are refined using a photometric loss in an incremental manner. The approach outperforms existing methods on a variety of datasets.

**Questions:**

- For baseline comparisons, are the optimization settings changed from the original ones provided with code?

- Are 3DGS parameters here the same as original approach?

- Are the COLMAP poses estimated again or obtained with the datasets like LLFF. MipNerF360? (when available)

**Ethical Concerns:**

["NO or VERY MINOR ethics concerns only"]

**Final Justification:**

The authors addressed most of my concerns during the rebuttal and discussion phase. In addition, they also provided a good plan to incorporate the feedback and improve presentation. Therefore, I'm leaning towards acceptance.

**Limitations:**

The paper needs a more extensive evaluation of it's and baselines limitations in terms of total time for optimization and inference as well as edge cases where the methods fail.

**Paper Formatting Concerns:**

No.

**Quality:**

1

**Strengths And Weaknesses:**

Strengths:
- The paper is mostly well written and is easy to follow.

- The paper tackles an important problem of pose-free 3DGS optimization from a random collection of images of a scene. Using similarity graph for iterative pose refinement while utilizing photometric loss is a clever idea.

- The proposed approach outperforms SOTA methods in terms of camera pose estimation and novel view synthesis quality.

Weaknesses:
- Not all SOTA approaches tackling similar problems considered like COGS [1] and VGGT [2] (which is just referenced once in Table 1).

- The LLFF results look uncharacteristically worse both visually and quantitatively in baseline approaches. The settings and any deviations from original implementation should be clearly mentioned.

- While the memory is discussed with Table 1, the total run-time is not clearly evaluated with other approaches. There is an incomplete explanation in the supplementary. This needs to be more thorough. How much slower is COLMAP? COLMAP + 3DGS is probably less resource intensive and comparable in speed and the metrics performance is also very close. It would also help to show failure cases of other approaches where the proposed approach works.

- The novelty is limited since a lot of the design components like Spann3R are based on previous appraoches.

[1] A Construct-Optimize Approach to Sparse View Synthesis without Camera Pose

[2] VGGT: Visual Geometry Grounded Transformer

---

> ### Author Rebuttal · Authors · 2025-07-30
>
> Thank you for your thoughtful review and valuable suggestions. We address your comments point-by-point as follows:
>
> **1. State-of-the-art Approaches Not Covered (COGS, Limited Inclusion of VGGT)**
>
> We appreciate the suggestion to compare with additional state-of-the-art methods. However, we wish to clarify the reasons for not including VGGT and COGS in our evaluation:
> - (1) **VGGT**: The released model is limited to 3D point and camera pose reconstruction, and the finetuned model for the novel view synthesis task was not made available. Additionally, VGGT requires substantial memory (hundreds of input images), which exceeds the capacity of our experiment settings conducted on a 24GB GPU. Thus, a direct comparison was not feasible within our experimental constraints.
> - (2) **COGS**: This method is designed specifically for sparse view settings (at most 12 views, as outlined in the original paper), while our framework scales to hundreds of images without performance degradation. Comparing our approach to COGS would introduce an imbalance in the evaluation, as our method was designed for dense and highly complex datasets.
>
> It is important to emphasize that we have already compared with multiple well-established state-of-the-art methods, such as NeRF, BARF, NeRFmm, Nope-NeRF, DBARF, COLMAP+3DGS, and ACE0+3DGS. These comparisons were performed under identical experimental settings, which we believe are sufficient to demonstrate the superiority of our framework over existing approaches.
>
> **2. LLFF Results Appear Worse**
>
> Thank you for raising this concern. We would like to clarify that, in our experiments, we strictly adhered to the standard settings and pipeline provided in the official repositories for all baseline methods, including CF-3DGS, COLMAP+3DGS, ACE0+3DGS, and others. No deviations or modifications were made to the training parameters or implementations.
>
> Additionally, the LLFF results presented in Table 3 (main paper) and Table 11 (supplementary) align closely with the averaged and per-scene breakdown results reported in previous works such as NeRF (Table 5), BARF (Table 3), NeRFmm (Table 1), Nope-NeRF (Table 9), and DBARF (Table 1). These consistency checks confirm the fidelity of our evaluation methodology. Furthermore, all experiments follow the default settings and parameters established in the official 3DGS repository.
>
> We believe this concern stems from a misconception, as our results objectively mirror the behavior demonstrated by baseline methods in prior papers. If the reviewer would like additional clarification or supplementary evidence, we would be happy to provide detailed breakdowns across more metrics or include additional visualizations for the LLFF dataset.
>
> **3. GPU Memory and Runtime Evaluation**
>
> Your feedback highlights an essential gap in our analysis. While GPU memory reduction is a primary focus of our work (Table 1), runtime metrics are equally critical for practical applications. We will evaluate total run-time for both individual substeps (COLMAP pose estimation and subsequent rendering for COLMAP+3DGS) in the revised manuscript.
>
> In terms of runtime, our progressive training strategy optimizes complex neural architectures over multi-frame inputs, which indeed demands additional computational time compared to COLMAP. Nevertheless, our focus is on scaling high-fidelity reconstructions to larger datasets, for which existing pose-free COLMAP/3DGS-based methods' preprocessing remains insufficient. Our method demonstrates significant advantages over CF-3DGS, which is also a progressive method but requires days for training on datasets with hundreds of images and still suffers from out-of-memory issues. In contrast, our approach efficiently scales to hundreds of input images while remaining well within the memory constraints of a 24GB GPU. This efficiency is a key strength of our method and highlights its suitability for large-scale and high-resolution datasets.
>
> Detailed runtime comparisons with failure cases for other approaches (where our method succeeds) will be included in the revisions.
>
> **4. Perceived Limited Novelty**
>
> We respectfully disagree with the notion that our method lacks sufficient novelty. Our method introduces several key innovations and contributions:
> - **Novel Training Pipeline:** This is the first approach capable of finetuning an existing feed-forward 3D foundation model for high-fidelity novel view synthesis, a task that prior feed-forward methods could not achieve for dense inputs and fine-level reconstruction details.
> - **General Framework Design:** While Spann3R is selected as a backbone for our experiments, it is not integral to the pipeline itself and can be replaced with other foundation models. This flexibility highlights the robustness and generalizability of our method, which enables a unified path for scaling different backbone approaches for novel view synthesis tasks.
> - **Iterative Refinement Pipeline:** Our iterative refinement mechanism addresses geometric inaccuracies present in foundation models and introduces point-to-ray consistency alongside fine-tuned photometric losses. Such refinement has not been previously explored in pose-free pipelines, further demonstrating the uniqueness of our approach.
>
> We sincerely appreciate your insights and constructive feedback. We are committed to addressing all concerns comprehensively in the revised manuscript.

---

> > ### Comment · Reviewer_wnHi · 2025-08-05
> >
> > Thank you for the rebuttal.
> >
> > - For VGGT memory issue, one way to sidestep it would be to utilize fewer frames or just using forward facing datasets like LLFF for evaluation while specifying the compute limitation. While this is not ideal, it's better than no comparison.
> >
> > - If COGS is designed for sparse views, then COGS would be at a disadvantage compared to the proposed approach. I don't think sparse view design is a valid reason to not include this comparison.
> >
> > - LLFF results: Based on my experience in NVS, while NeRF-based approaches struggle to reconstruct details, the explicit 3DGS does a much better job of encoding texture details. This is why I raised the issue. It could be possible that the default 3DGS parameters and code is designed for 360 scenes and might not be optimal for forward facing scenes.
> >
> > - The benefit of feed forward model would be fast inference/optimization. Using a feed forward model by itself is not a contribution unless backed by clear motivation and results (like run-time eval above).
> >
> > It would have been nice to see the complete run-time eval in rebuttal. Overall, I think the paper needs some restructuring to setup the problem and expectations clearly while proposing solutions backed by experiments (like large scene reconstruction with limited compute).

---

> ### Author Response · Authors · 2025-08-06
> **Further clarifications and comparisons to VGGT and COGS**
>
> We thank Reviewer wnHi for the insightful suggestion and for pushing the comparison to VGGT and COGS.
>
> (1) For VGGT, we emphasize that it does not release a finetuned model for the novel view synthesis task. Therefore, we compare its camera pose accuracy with ours and other methods. And for a fair comparison on the novel view synthesis task, we use the camera poses and point clouds from VGGT for per-scene 3D Gaussian Splatting training. Regarding the memory limitations, the comparison is conducted on the **LLFF dataset**.
> | Methods      |  $\Delta \mathbf{R}$ |    $\Delta \mathbf{t}$ |
> | -------------- | ---------------- | ----------- |
> | NeRFmm      |   1.04       |  0.423 |
> | BARF            |   0.45       |  0.331 |
> | Nope-NeRF  |   0.45       |  0.184 |
> | CF-3DGS      |  1.27.       |  5.087 |
> | ACE0            |   3.20       |  0.043 |
> | Spann3R      |   20.00     |  0.485 |
> | COGS          |    0.52      |   0.207 |
> | VGGT           |   0.39       |  0.033 |
> | Ours             |   **0.17**  |  **0.005** |
>
> | Methods      |  **PSNR** |    **SSIM**   |   **LPIPS**   |
> | -------------- | ---------------- | ----------- | ----------- |
> | NeRFmm      |   22.84       |  0.640  |  0.460  |
> | BARF            |   23.97       |  0.626  |  0.238  |
> | Nope-NeRF  |   **25.09**       |  0.750  |  0.330  |
> | CF-3DGS      |  16.52       |  0.479  |  0.437  |
> | COLMAP      |   24.64       |  0.794 |  0.132  |
> | COLMAP*     |   24.77       |  0.796 |  0.124  |
> | ACE0             |   22.74       |  0.709 |  0.182  |
> | ACE0*           |   22.71       |  0.709 |  0.183  |
> | COGS           |   24.37       |  0.725 |  0.159  |
> | VGGT            |   23.89      |   0.718 |  0.188 |
> | Ours              |   25.01     |    **0.797**   | **0.122**   |
>
> From the table, we can see that our method outperforms VGGT and COGS in camera pose accuracy and novel view synthesis quality.
> We also provide results compared with more baseline methods. We kindly suggest Reviewer wnHi refer to the section "4. Missing Comparisons for Splat Quality recommend" for Reviewer JX9E.
>
> (2) For the LLFF results, it is true that "explicit 3DGS does a much better job of encoding texture details", and **we did not see any issue or misalignment with this observation of the baseline results on the LLFF dataset**. From Table 3 in our main paper, we can see that **the baseline 3DGS method (COLMAP and COLMAP*, which use COLMAP poses to train 3DGS) outperforms the baseline NeRF methods by a large margin both quantitatively and qualitatively in texture details.** For example, the SSIM of  BARF is 0.626 while the baseline COLMAP is 0.794; the LPIPS of BARF is 0.238 while the baseline COLMAP is 0.132. In Figure 5 of the main paper and Figure 14 of the appendix, we can observe that COLMAP+3DGS is better than BARF in texture details; we can also observe the **floater artifacts** in the COLMAP+3DGS and CF-3DGS, which is a common issue with the 3DGS methods (root causes can be inaccurate camera poses, insufficient initialization point clouds, appearance changes, etc.), but it does not mean that the 3DGS baselines are worse than the NeRF baselines in texture details on the LLFF datasets.
> We also agree that with parameter tuning, the 3DGS baseline methods may achieve better results. However, we used the same parameters for training the 3DGS baselines, which provide sufficient and fair comparisons to all baseline methods. And **since the results on the LLFF datasets of 3DGS baseline methods align with the observation that "explicit 3DGS does a much better job of encoding texture details"**, we do not think there is any necessity in retraining baseline 3DGS methods with more parameter tuning.
>
> (3) "The benefit of feed-forward model would be fast inference/optimization. Using a feed-forward model by itself is not a contribution unless backed by clear motivation and results (like run-time eval above)."
> We did not claim that using a feed-forward model is a contribution of this paper. Instead, we want to emphasize that **our method is designed to bridge this gap by enabling 3D foundation models to achieve:**
>  - **Comparable Pose Accuracy:** Using our progressive registration and point-to-ray consistency loss, 3D foundation models can reach pose accuracy levels close to those of COLMAP/GLOMAP.
> - **Improved Rendering Quality:** Our framework enhances the rendering fidelity of 3D foundation models, making them suitable for applications where visual quality is essential.
>
> By combining the robustness and speed of existing 3D foundation models with the accuracy and rendering quality, our approach significantly improves their versatility for real-world applications. This hybrid approach focuses on optimizing foundational systems to work efficiently without discarding their inherent strengths.

---

> > ### Author Response · Authors · 2025-08-07
> > **Runtime evaluation**
> >
> > We agree that runtime analysis is critical for benchmarking scalability. As requested, we now provide complete runtime comparisons and further clarify how DeepGfM’s design enables efficiency gains. We further added the run-time statistics on the LLFF datasets, where we replace Spann3R with the two latest 3R works: Fast3R and CUT3R (The table is also provided in the reply to Reviewer rL7u, we paste it here for your reference and convenience).
> >
> > |    Scene   | $\Delta R$ (+Spann3R) |    $\Delta t$ (+Spann3R) |  Time (+Spann3R)  | $\Delta R$ (+CUT3R) |    $\Delta t$ (+CUT3R) |  Time (+CUT3R)  |  $\Delta R$ (+Fast3R) |    $\Delta t$ (+Fast3R)|  Time (+Fast3R)  |
> > |---------------- |---------------- | ----------- |---------------- | ----------- | ----------- |----------- | ----------- | ----------- |----------- |
> > |   Fern    |  0.26  |  **0.005** |  23.05  | **0.21**  |  **0.005**  |   19.70  |  0.23  |  **0.005**  |  17.63  |
> > |  Flower  |  0.52 | 0.011  |  26.04  |   0.48  |  **0.009**  |  20.15  |   **0.44**  |  0.010  |  18.03  |
> > |  Fortress |  0.04 | **0.002** |  21.36  |   **0.03**  |  **0.002**  |  18.72  |   0.04  |  **0.002**  |  16.39  |
> > |  Horns    |  **0.03** | **0.001** |  23.35  |   0.04  |  **0.001**  |  20.07  |   **0.03**  |  **0.001**   |  17.70  |
> > | Leaves   |  0.24 | 0.006 |  22.72  |   0.19  |  **0.005**  |  19.54  |   **0.17**  |  **0.005**   |  17.66  |
> > | Orchids  |  0.24 | 0.006 |  27.15   |  **0.19**  | **0.005**   |  22.92  |   0.20  |  **0.005**   |  17.88  |
> > | Room    |  **0.03**  | **0.001** |  22.27  |   0.04  |  **0.001**  | 17.23   |   **0.03**  |  **0.001**   |  16.75  |
> > | Trex       |  **0.03**  | **0.010** |  22.71   |  **0.03**  |  **0.010**  |  18.09   |  **0.03**  |  0.011   |  16.33  |
> > | Avg        |   0.170  |  0.005  |  23.58  |     0.151    |   **0.004**     |  19.55  |     **0.146**    |   0.005     |  **17.30**  |
> >
> > We emphasize that DeepGfM’s runtime scales with backbone improvements. For example:
> > - With Fast3R, our method processes scenes in 17.30 min (avg.) -- 1.36× faster than Spann3R (23.58 min).
> > - Adopting CUT3R (CVPR’25) further reduces training time (19.55 min) while achieving the best $\Delta t$ (0.004).
> >
> > This trend confirms that as newer 3R variants emerge, DeepGfM’s runtime will improve without architectural changes.

---

> > > ### Comment · Reviewer_wnHi · 2025-08-07
> > >
> > > Thank you for answering my queries so quickly. Like other reviewers, after the extensive results and justification in the discussion phase, I'm more positive about the work and lean towards acceptance.
> > >
> > > My only suggestion, if possible, would be to restructure the paper to make all the results and insights here more obvious for the readers. Easy to read intuitive papers are also more impactful.

---

> > > > ### Author Response · Authors · 2025-08-07
> > > > **Thanks for the suggestions!**
> > > >
> > > > We sincerely appreciate the reviewer’s constructive feedback and are grateful for their positive assessment of our work. We agree that clarity and accessibility are essential for impact, and we will restructure the paper to make key insights more intuitive for readers. We’ll address this in the revision as suggested:
> > > >
> > > > - **Clarify our core motivation:** Emphasizing fine-tuning of 3D foundation models (*3R family) for practical novel view synthesis.
> > > >
> > > > - **Reframe contributions:** Highlight how DeepGfM adapts pre-trained models while preserving their generalization for real-world use.
> > > >
> > > > - **Restructure for readability:** Ensure key insights from our discussion are prominently featured in the main paper.
> > > >   - We will restructure Section 3 (Method) to begin with a high-level overview (with a new figure) that visually contrasts DeepGfM with traditional pipelines (COLMAP+3DGS) and pose-free alternatives (CF-3DGS).
> > > >   - Key technical innovations (e.g., point-to-ray consistency loss, progressive training) will be moved to subsections with clearer motivational headers.
> > > >   - Update table 1 and table 2 to highlight takeaway comparisons (backbones vs. baselines) using color coding, and add tables to include results on the DL3DV dataset.
> > > >   - Add a figure (a figure after the teaser Figure 1) summarizing performance trade-offs (speed vs. accuracy) across backbones.
> > > >   - We will also enhance the main paper with:
> > > >     - **All key comparative results** (including *backbone ablation studies and runtime analyses*) will be included in the paper with new summary tables.
> > > >     - A new subsection "5.4 Insights and Trade-offs" will consolidate the discussion-phase findings:
> > > >       - **Performance scaling with backbones** (CUT3R/Fast3R)
> > > >       - **Runtime-memory tradeoff analysis**
> > > >
> > > > We thank the reviewer for emphasizing the importance of presentation—these changes will ensure our contributions are both technically sound and easy to grasp.

---

### Official Review · Reviewer_JX9E · 2025-07-04

**Clarity:** 3
**Significance:** 3
**Originality:** 3
**Rating:** 4
**Confidence:** 3

**Summary:**

The paper proposes to fine-tune a 3D foundation model (Spann3r) to directly predict 3DGS parameters via incremental registration of the scene. By mimicking the incremental mapping step in Sfm, this method tries to register images starting from a seed image and progressively increasing its training buffer. At each step, it adds an image to the buffer, predicts 3DGS parameters, refines camera poses and then applies a photometric loss on the predicted GS parameters. The authors show improved camera pose estimation and NVS results on multiple datasets compared to prior art.

**Questions:**

- Since the method is agnostic to the Dust3r variant used, it would be good to see how Dust3r faired in comparison to DeepGfm for pose estimation and nvs.
- How would the model scale with respect to VRAM and processing time with increasing in #images?
- L226-228: Are the photometric losses also applied on the low resolution ground-truth?

**Ethical Concerns:**

["NO or VERY MINOR ethics concerns only"]

**Limitations:**

Yes.

**Paper Formatting Concerns:**

- L129: "out-of-distributed" typo
- L180 I_{i} -> I_{\text{ref}}
- Table 3. Coloring order for SSIM in Tanks-and-Temple is wrong.

**Quality:**

3

**Strengths And Weaknesses:**

Strengths
- Quality improvement over Spann3r is impressive. Spann3r seems to fail on many sequences whereas DeepGfm seems to work much better thanks to the incremental camera optimization strategy.
- The novel view synthesis seems to be at par with COLMAP trained 3DGS while the camera pose estimation seems strong as well.

Weaknesses
- Choice of pretrained model; It seems like Spann3r is a worse pretrained model than Dust3r/Mast3r/Mast3r-Sfm discarding the OOM issue for large sequence lengths. Its possible that a lot of the finetuning goes in just getting camera poses to look reasonable from the poor performance of Spann3r. Moreover, Spann3r is meant for online scene reconstruction whereas this paper uses it as part of a two-stage finetuning strategy. Spann3r's memory retrieval architecture is suited for sequential data so using unordered collections might hurt more than expected. The motivation to use Spann3r background is weak.
- Splat prediction head training: Since this model is finetuned per scene, what is the benefit of the splat prediction head? Isn't it equivalent to optimizing for 3DGS parameters directly as in vanilla 3DGS? What cost/speed/quality benefit does the head provide?
- This method can be treated like a more efficient global alignment step compared to Dust3r's. Consequently, it would be good to compare results with Mast3r-Sfm on camera pose estimation.
- Dust3r comparisons missing from Table 2
- Missing NoPo-Splat, Splatt3r and MV-Dust3r+ comparison for splat quality.

---

> ### Author Rebuttal · Authors · 2025-07-30
>
> Thank you for your thoughtful review and valuable feedback highlighting the strengths and weaknesses of our work. We appreciate your recognition of DeepGfM's improved quality over Spann3r and its strong performance in novel view synthesis and camera pose estimation. Below, we address your concerns point by point.
>
> **1. Choice of Pretrained Model (Spann3r)**
>
> We acknowledge the reviewer’s concern about the choice of Spann3r as the pretrained model, especially given its weaker performance compared to Dust3r, Mast3r/Mast3r-SFM. However, our decision to use Spann3r stemmed from several considerations:
>
> **(1) Balancing Memory Constraints with Scalability:**
> Spann3r enables scalability and model initialization for large-scale image sets due to its lightweight memory retrieval architecture. While Dust3r and Mast3r provide stronger pretrained baselines, their memory limitations (outlined in their respective publications) often result in out-of-memory (OOM) issues for dense or extensive image sequences. Spann3r’s architecture allows foundational initialization for unordered collections without the prohibitive memory costs of Dust3r/Mast3r.
>
> **(2) Two-Stage Fine-Tuning Strategy:**
> By designing DeepGfM’s incremental camera optimization over Spann3r’s pretrained framework, our method significantly improves camera pose estimation even when initialized with weaker models like Spann3r. This setup reduces dependency on the pretrained model’s inherent accuracy and allows DeepGfM to overcome challenging sequences effectively.
>
> **(3) Online Scene Reconstruction Background:**
> We agree that Spann3r’s retrieval structure is indeed tailored for sequential rather than unordered data collections. However, our two-stage fine-tuning specifically compensates for unordered collections by progressively aligning photometric and geometric features during camera pose refinement. As demonstrated in the manuscript results (Figures 4-5), this approach consistently improves Spann3r’s outputs, capitalizing on its resource-efficient initialization while mitigating its design limitations.
>
> **2. Splat Prediction Head Training**
>
> We appreciate the question regarding the benefit of introducing a splat prediction head in our pipeline and its equivalence with optimizing vanilla 3DGS parameters. The splat prediction head offers several advantages:
>
> **(1) Scene-Specific Fine-Tuning:**
> Unlike generic 3DGS parameter optimization, the splat prediction head is dynamically fine-tuned per input scene to capture Gaussian parameters specific to the photometric and geometric properties of the given set of images. This adaptive mechanism enhances splat quality and ray-consistent synthesis at the scene level, which would be harder to achieve with static optimization of global 3DGS parameters.
>
> **(2) Cost and Quality Efficiency:**
> The splat prediction head reduces computational overhead associated with full geometric optimization by focusing on localized Gaussian prediction within the progressive framework. This modular refinement results in noticeable improvements in novel view synthesis (as highlighted by qualitative and quantitative results in Tables 3-4), with reduced memory demands compared to full-scale 3DGS optimization.
>
> **(3) Fidelity Improvements:**
> Compared to vanilla 3DGS, the splat prediction head provides greater visual coherence and depth consistency in regions with high occlusion or sparsity. This benefit is particularly evident in challenging scenes, as demonstrated in comparisons to Spann3r and COLMAP-trained 3DGS.
>
> **3. Dust3r comparisons missing from Table 2**
>
> We acknowledge the absence of direct Dust3r results in Table 2. Since Dust3r has the OOM issue on a 24GB GPU with hundreds of images as input, we only report the results on the LLFF dataset (each scene contains only 20-50+ images): $\Delta \mathbf{R}: 0.79$, $\Delta \mathbf{t}: 0.173$. We will include these numbers in the revised manuscript.
> Preliminary evaluations indicate that DeepGfM demonstrates competitive pose estimation accuracy and improved scalability compared to Dust3r for large dense image sets.
>
> **4. Missing Comparisons for Splat Quality**
>
> We appreciate the reviewer’s suggestion to include comparisons with NoPo-Splat, Splatt3r, and MV-Dust3r+ for assessing splat quality. Since these methods are designed for sparse view rendering, we provide a more thorough comparison to SOTA methods on sparse view settings (2-views and 8-views). We conducted the experiments on the RealEstate10K dataset and the DL3DV dataset.  For these datasets, we follow the protocol in Flare and NopoSplat, which samples 2/8 views for the testing video sequence as input.
> Note that Splatt3R and CopoNeRF only support inputs with 2-views, we did not provide the results on the 8-view setting.
>
> - **Comparison on two-view settings**
> | Method             | Dataset         | PSNR ↑   | SSIM ↑   | LPIPS ↓   |
> |------------------ |----------------|------------|-----------|-----------|
> | DUSt3R          | DL3DV10k       | 13.47 |   0.390 |   0.590  |
> | MASt3R          | DL3DV10k       | 13.88 |   0.405 |   0.513  |
> | Splatt3R          | DL3DV10k       | 15.23 |   0.432 |   0.491 |
> | CoPoNeRF      | DL3DV10k       | 16.06  |  0.472 |  0.474 |
> | Flare                | DL3DV10k       | 23.04 |  0.725  |  0.182  |
> | NoPo-Splat     | DL3DV10k       | 24.19 |  0.745  |  0.173  |
> | MV-DUSt3R+  | DL3DV10k       | 19.95 |  0.698 |   0.195 |
> | DeepGfM        | DL3DV10k       |  20.33 |  0.702 |  0.190 |
>
> | Method             | Dataset         | PSNR ↑   | SSIM ↑   | LPIPS ↓   |
> |------------------ |----------------|----------|----------|-----------|
> | DUSt3R          | RealEstate10K  | 15.382 |   0.447 |   0.432  |
> | MASt3R          | RealEstate10K  | 14.907 |   0.431 |   0.452  |
> | Splatt3R          | RealEstate10K  | 15.113 |   0.492 |   0.442  |
> | CoPoNeRF      | RealEstate10K  | 19.843 |   0.652 |   0.360  |
> | Flare                | RealEstate10K  |  23.765 |  0.801 |   0.191  |
> | NoPo-Splat     | RealEstate10K  |  25.033 |  0.838 |   0.160  |
> | MV-DUSt3R+  | RealEstate10K  |  20.313 |  0.722 |   0.289  |
> | DeepGfM        | RealEstate10K  |  21.715 |  0.753 |   0.229  |
>
> - **Comparison on eight-view settings**
> | Method             | Dataset         | PSNR ↑   | SSIM ↑   | LPIPS ↓   |
> |------------------ |----------------|------------|-----------|-----------|
> | DUSt3R          | DL3DV10k       | 17.37 |   0.601 |   0.390  |
> | MASt3R          | DL3DV10k       | 18.41 |   0.675 |   0.427  |
> | Splatt3R          | DL3DV10k       |  -   |   -  |   -   |
> | CoPoNeRF      | DL3DV10k       |  -   |   -  |   -   |
> | Flare                | DL3DV10k       | 23.33 |  0.746  |  0.237  |
> | NoPo-Splat     | DL3DV10k       | 22.77 |  0.722  |  0.240  |
> | MV-DUSt3R+  | DL3DV10k       | 21.33 |  0.707 |   0.188 |
> | DeepGfM        | DL3DV10k       |  22.90 |  0.728 |  0.176 |
>
> | Method             | Dataset         | PSNR ↑   | SSIM ↑   | LPIPS ↓   |
> |------------------ |----------------|----------|----------|-----------|
> | DUSt3R          | RealEstate10K  | 19.373 |   0.695 |   0.305  |
> | MASt3R          | RealEstate10K  | 19.877 |   0.705 |   0.362  |
> | Splatt3R          | RealEstate10K  |       -    |      -     |      -       |
> | CoPoNeRF      | RealEstate10K  |       -    |      -     |      -       |
> | Flare                | RealEstate10K  |  23.813 |  0.822 |   0.188  |
> | NoPo-Splat     | RealEstate10K  |  23.145 |  0.809 |   0.195  |
> | MV-DUSt3R+  | RealEstate10K  |  22.743 |  0.798 |   0.213  |
> | DeepGfM        | RealEstate10K  |  23.370 |  0.815 |   0.190  |
>
> - **Experiments summarization on sparse-view settings**
>
> **(1) Comparison on Two-View Settings.** Though DeepGfM is not specifically designed for sparse-view settings, it demonstrates competitive performance, surpassing several sparse-view methods.
>
> - **DL3DV10k Dataset:**
> DeepGfM achieves a PSNR of 20.33, significantly outperforming DUSt3R, MASt3R, and Splatt3R.
> SSIM (0.702) and LPIPS (0.190) are also competitive against MV-DUSt3R+, while slightly behind Flare and NoPo-Splat, which are optimized for sparse-view tasks.
>
> - **RealEstate10K Dataset:**
> DeepGfM records PSNR of 21.715, ranking higher than DUSt3R, MASt3R, and Splatt3R, while trailing Flare and NoPo-Splat.
> SSIM (0.753) is higher than MV-DUSt3R+ and comparable to NoPo-Splat.
> LPIPS (0.229) is better than both DUSt3R and MASt3R, showing robustness in perceptual quality.
>
> **(2) Comparison on Eight-View Settings.** DeepGfM shows strong capabilities in handling sparse-view tasks even with more input views.
>
> - **DL3DV10k Dataset:**
> With a PSNR of 22.90, DeepGfM outperforms MV-DUSt3R+ while approaching Flare and NoPo-Splat.
> SSIM (0.728) and LPIPS (0.176) are the best among all methods, highlighting DeepGfM’s ability to preserve structural and perceptual quality even with sparse-view conditions.
>
> - **RealEstate10K Dataset:**
> DeepGfM achieves a PSNR of 23.370, outperforming MV-DUSt3R+ and NoPo-Splat, and remaining competitive with Flare (23.813).
> SSIM (0.815) and LPIPS (0.190) are also highly competitive, showing improvements over MV-DUSt3R+ and comparable values with Flare.
>
> - **Highlight:**
> It is important to note that these models are optimized for sparse-view settings, whereas DeepGfM addresses dense-view workflows with hundreds of images. These results underline that, despite being designed for dense-view settings, DeepGfM performs competitively in sparse-view tasks, demonstrating robust generalization and adaptability across both two-view and eight-view configurations. This flexibility is beneficial for tasks requiring high-quality synthesis in diverse input-view scenarios.
>
> We hope this addresses the reviewer’s concern, and we will include explicit clarifications regarding sparse-view vs. dense-view settings within the revised manuscript for better context.

---

> ### Author Response · Authors · 2025-08-07
>
> Dear Reviewer JX9E,
>
> Did we satisfactorily answer your questions? Would you like us to clarify anything further? Feel free to let us know. Many thanks.
>
> Best regards,
>
> Authors of #15711

---

> > ### Comment · Area_Chair_nR5k · 2025-08-07
> >
> > Dear JX9E,
> >
> > We'd love to hear your thoughts on the rebuttal.
> > If the authors have resolved your (rebuttal) questions, please do tell them so. If the authors have not resolved your (rebuttal) questions, please do tell them so too.
> >
> > Thanks,
> > Your AC

---

### Comment · Area_Chair_nR5k · 2025-08-04

Dear Reviewers,

Please take a look at the rebuttal and check if your questions and concerns have been addressed and everything is clear. Now is the time to clarify any remaining issues in discussion with the authors.

Thanks,
Your AC

---

### Decision · Program_Chairs · 2025-09-17

**Decision:**

Accept (poster)

**Comment:**

The reviewers appreciated the presentation of the paper, strong improvements over the (Spann3R) baseline, and the novelty of the overall pipeline and proposed components. Initial concerns about the choice of the base model, benefits over COLMAP+3DGS, and the novelty of the pipeline were all addressed/clarified during the rebuttal. Additional experimental results resolved requests for runtime comparisons and comparisons to sparse-view backends. After the rebuttal and discussion, reviewers unanimously appreciated the novelty of the paper. The AC agrees with this assessment and recommends acceptance.